# Eating Disorder Risk Among Adolescents: The Influence of Dietary Patterns, Physical Activity, and BMI

**DOI:** 10.3390/nu17061067

**Published:** 2025-03-19

**Authors:** Anca Georgiana Ispas, Alina Ioana Forray, Alexandra Lacurezeanu, Dumitru Petreuș, Laura Ioana Gavrilaș, Răzvan Mircea Cherecheș

**Affiliations:** 1Department of Public Health, College of Political, Administrative and Communication Sciences, Babes-Bolyai University, 400376 Cluj-Napoca, Romania; georgiana.ispas@publichealth.ro (A.G.I.);; 2Asociația Wello, 400686 Cluj-Napoca, Romania; 3Discipline of Public Health and Management, Department of Community Medicine, “Iuliu Hațieganu” University of Medicine and Pharmacy, 400012 Cluj-Napoca, Romania; 4Faculty of Food Science and Technology, University of Agricultural Sciences and Veterinary Medicine, 400372 Cluj-Napoca, Romania; 5Department 2, Faculty of Nursing and Health Sciences, “Iuliu Hațieganu” University of Medicine and Pharmacy, 400012 Cluj-Napoca, Romania

**Keywords:** eating disorders, adolescent, body mass index, dietary habits, physical activity, processed foods, mediterranean diet, body image, mental health, health promotion, weight perception, Romania

## Abstract

Background/Objectives: Eating disorders (EDs) are rising among adolescents. BMI, diet, and physical activity influence ED risk, but their interactions are complex and vary by population. Considering sociocultural changes, this study investigated the prevalence and risk factors for EDs in Romanian adolescents. We evaluated demographic factors, BMI, dietary behaviors, physical activity (PA), and previous specialist consultations using the Eating Attitudes Test-26 (EAT-26) and assessed the psychometric properties of the EAT-26 in this population. Methods: A cross-sectional study included 423 youths aged 13 to 20 from Cluj-Napoca. Participants completed an online survey on demographics, BMI, diet, physical activity, and previous health consultations. ED risk was assessed using the EAT-26. Confirmatory and exploratory factor analyses were conducted on the EAT-26, and binary logistic regression identified predictors of ED risk. Results: The prevalence of ED risk was 26.5%, with females having 1.61 times higher odds than males. Consulting a weight or mental health specialist increased the odds of increased ED risk (OR = 3.76). Higher BMI showed a trend in the unadjusted model (OR = 1.06). An initial CFA of the EAT-26 showed a suboptimal fit. Over 60% of the sample reported frequent consumption of ultra-processed foods. Conclusions: Our findings highlight a significant ED risk in Romanian youth, filling a critical gap in regional public health evidence. Despite limitations due to the cross-sectional design and self-reported data, the results stress the need for measures promoting balanced diets, positive body image, and improved mental health access to reduce the growing prevalence of adolescent ED risk in Romania.

## 1. Introduction

EDs are a pervasive global health challenge, affecting diverse populations and regions. The prevalence of EDs and associated disability-adjusted life years (DALYs) has seen a significant increase worldwide between 1990 and 2017, underscoring their escalating public health impact [1]. Based on the Global Burden of Disease Study 2019 data (reflecting estimates from 2018 onward), the global prevalence of EDs among children and adolescents is around 137 per 100,000. Males have a rate of approximately 90 per 100,000, while females are at about 188 per 100,000 [2]. Specifically, anorexia nervosa affects roughly 61 per 100,000, and bulimia nervosa affects about 79 per 100,000. EDs, including anorexia nervosa, bulimia nervosa, and binge eating disorder (BED), are associated with high mortality rates, reduced quality of life, increased healthcare costs, and adverse long-term health consequences [3]. Adolescents are particularly vulnerable due to the psychological, physiological, and social changes they experience during puberty, making early detection and intervention crucial [4]. BED, characterized by recurrent episodes of overeating, has a global prevalence of 0.9% and is often underdiagnosed due to cultural and systemic barriers, particularly in low- and middle-income countries [5].

Despite the growing body of international research on EDs, data specific to Romanian populations, particularly adolescents, remain scarce and often dated. While studies conducted in 2007 [6], 2009 [7], and 2010 [8] shed light on ED prevalence among high school students—reporting figures such as 0.6% for anorexia nervosa and 1.3% for bulimia nervosa—more contemporary data focusing on younger cohorts are limited. Although a 2019 study [9] highlighted ED concerns among Romanian university students (prevalence of approximately 21.8% in males and 24.7% in females) and another study from 2024 involving Romanian medical students suggested that 37.1% of students are at risk of eating disorders [10], adolescent-specific data remain underrepresented, thereby limiting our understanding of the developmental trajectory and contextual factors influencing ED onset. The paucity of nationally representative, adolescent-focused research underscores the critical need for up-to-date, large-scale studies to inform targeted prevention and intervention strategies within Romania.

The global obesity epidemic adds a layer of complexity to the issue of EDs. Common risk factors such as body dissatisfaction, weight stigma, and unhealthy eating behaviors create a link between obesity and EDs. This overlap exacerbates physical and mental health problems, underscoring the urgent need for integrated prevention strategies that target both conditions [11,12]. Weight stigma continues to lead to adverse outcomes for individuals with obesity and EDs, highlighting the importance of a holistic treatment approach and training compassion for these conditions [13,14].

EDs arise from diverse biological, behavioral, and sociocultural influences. Among these, BMI is a complex factor. While low premorbid BMI is often observed in anorexia nervosa, the disorder frequently begins with intentional weight loss due to perceived or actual excess weight, often influenced by peer and societal pressures. Conversely, elevated BMI is correlated with binge eating and bulimia nervosa [15]. Longitudinal studies suggest that childhood BMI trajectories may predict future ED diagnoses, highlighting the importance of early detection and intervention [16], with lower BMI patterns often preceding anorexia nervosa and higher BMI trajectories correlating with bulimia nervosa and binge eating disorder [15]. This association likely reflects the interplay between body image concerns, weight-related teasing, and dieting behaviors, which can amplify vulnerability to disordered eating in adolescence [17,18]. Consequently, early detection and intervention strategies that address both physiological and psychosocial risk factors are crucial for preventing or mitigating ED onset. Several psychosocial factors, including body dissatisfaction, depressive symptoms, and pubertal changes, mediate the association between BMI and EDs. Additionally, traumas such as bullying further increase the risk of ED, particularly in adolescents with a higher BMI [19]. These results highlight the complex interplay between physical and mental health in the development of ED susceptibility.

While BMI is an important consideration, research suggests that dietary behaviors may play an even more direct role in ED risk. The type and quality of food consumed can influence both physiological and psychological aspects of disordered eating, making dietary habits a crucial modifiable factor in adolescent health. Healthy habits such as regular breakfast and family meals protect against overweight and obesity and reduce associated ED risks [20,21]. Conversely, consumption of highly processed foods (UPFs), characterized by high levels of sugar, unhealthy fats, and additives, worsens emotional eating and metabolic disorders and increases the risk of EDs [22]. Adolescents exposed to UPFs report lower body satisfaction and increased emotional distress, creating a feedback loop that increases vulnerability to ED [23]. On the other hand, the Mediterranean diet (MedDiet), which focuses on whole foods, unsaturated fats, and polyphenols, has shown promise as a dietary intervention for vulnerable populations. Adherence to MedDiet has been associated with reductions in depression and weight, suggesting its potential to mitigate the risk of EDs [24,25]. Furthermore, moderate to vigorous PA can improve cardiorespiratory fitness, mental health, and metabolic outcomes while reducing the lifetime risk of depression [26,27]. However, excessive or compulsive exercise is associated with an increased risk of ED, particularly in sports environments, where weight and aesthetics are a priority [28]. The prevalence of exercise addiction risk ranges from 1% to 52%, increasing to 80% when co-occurring with eating disturbances [29]. Additionally, increased screen time—particularly exposure to social media—has been linked to body dissatisfaction and a higher risk of disordered eating behaviors, emphasizing the need to consider digital exposure as a contributing factor [30].

Romania offers a unique context for the study of EDs due to ongoing socioeconomic and cultural changes. Positive correlations between age, BMI, and physical inactivity among Romanian adolescents highlight the need for targeted interventions to address these vulnerabilities [31]. Emotional eating, particularly in older adults, highlights the importance of early emotional regulation strategies to prevent long-term health consequences [32]. Validation of the Dutch Eating Behavior Questionnaire in Romania highlights the importance of culturally adapted tools to assess ED risks in different populations [33]. Romanian adolescents exhibit distinct dietary behaviors shaped by a combination of traditional food consumption patterns and modern dietary influences. Studies indicate that Romanian youth show lower adherence to the Mediterranean Diet than their peers in Italy, with a higher intake of dairy products, nuts, and home-cooked meals, yet lower consumption of fast food and sweets [34]. Post-pandemic trends have led to a decline in diet quality among Romanian adolescents, with only 20.7% maintaining a healthy diet. There has been a decrease in fruit and vegetable intake and an increase in sedentary behaviors, accompanied by a rise in the consumption of energy-dense, nutrient-poor foods like fast food and sugary beverages [35].

The Eating Attitudes Test-26 (EAT-26) is a widely used, 26-item questionnaire designed to screen for eating disorder risk, with scores above 20 typically suggesting a need for further clinical evaluation [36]. It assesses symptoms and concerns related to anorexia nervosa and bulimia nervosa, measuring dimensions like dieting, bulimia, and oral control [36]. However, its reliability and validity can vary across different populations, particularly among adolescents. Studies in various countries, including in Ireland [37], a general review highlighting cross-cultural issues [38], and mainland China [39], have shown that the EAT-26’s original factor structure may not always hold, and optimal cutoff scores can differ. Therefore, assessing the EAT-26’s psychometric properties within a specific adolescent population, like this Romanian cohort, is crucial to ensure its accurate and effective use as a screening tool and to determine if cultural adaptations are needed. Furthermore, some studies consider the EAT-26 a reliable and valid screening tool. However, its limitations, including a tendency to overestimate eating disorder (ED) risks and issues with scale functionality, underscore the importance of using complementary assessment tools to ensure a more accurate and comprehensive evaluation of ED symptoms and related behaviors [40].

The relationship between BMI, dietary patterns, and PA in adolescent ED risk is still poorly understood despite the expanding corpus of research on EDs, especially in Eastern European populations. Western cultures, where body image ideals are often driven by social media and widespread promotion of thinness, tend to report higher levels of body dissatisfaction and restrictive eating behaviors [41,42]. In contrast, Eastern European cultural traditions, which emphasize familial meal structures and home-prepared foods, may exert a protective effect against certain ED risk factors but could also encourage emotional eating patterns [43]. Additionally, research highlights disparities in nutritional education and awareness of sustainable diet practices, with Romanians needing greater access to health-conscious dietary interventions [44]. These findings underscore the importance of targeted public health efforts to promote balanced nutrition and ED prevention strategies among Romanian youth. Given the scarcity of research on this topic in Romanian samples, more research should further explore the interplay between diet quality, cultural influences, and ED risk factors.

This study aimed to comprehensively investigate the prevalence and multifaceted determinants of ED risk among a cohort of adolescents in Cluj-Napoca, Romania. Specifically, we sought to determine the relative contributions of demographic factors (gender, age, residence), anthropometric measures (BMI), dietary behaviors (adherence to the Med Diet and ultra-processed food (UPF) consumption), and PA levels to the likelihood of exhibiting ED symptoms, as measured by the EAT-26. Furthermore, the research explored the potential relationship between consulting a specialist and the EAT-26 score. A secondary aim was to validate the psychometric properties of the EAT-26 within this specific Romanian adolescent population, given the potential for cultural variations in the manifestation and reporting of ED symptoms.

## 2. Materials and Methods

### 2.1. Study Design and Participants

This study employed a cross-sectional design to investigate the risk of EDs among adolescents and the roles of BMI, diet, PA, and knowledge of diet culture. Data were collected from a convenience sample of 423 adolescent participants attending seven high schools in Cluj-Napoca, Romania: George Coșbuc, Alexandru Borza, Ana Aslan, Augustin Maior, Nicolae Bălcescu, Spiru Haret, and Spectrum. Teachers at the participating high schools invited students to participate in their classes. All students who expressed interest and returned the necessary consent/assent forms were included in the study. Inclusion criteria were enrollment at one of the participating high schools and willingness to participate, as evidenced by the return of signed consent/assent forms. There were no specific exclusion criteria. All students who showed interest and returned the necessary consent and assent forms were eligible for the study, provided they (1) were enrolled in one of the seven participating high schools, (2) submitted signed parental or guardian consent (for minors) along with personal assent, and (3) had no clear signs of psychological or learning disabilities. Therefore, the final sample size included all 423 respondents.

To assess representativeness, we compared our sample distribution with official 2024 data from the National Institute of Statistics. The total adolescent population (ages 13–20) in Cluj County is 53,898, with 48.9% female and 51.1% male, and a rural-to-urban distribution of 42.8% to 57.2%, respectively. Our study sample closely mirrors these demographics, with 52.7% female and 47.3% male participants and a rural-to-urban distribution of 27.7% to 72.3%. Additionally, a sample size calculation indicated that at least 382 participants were required for a 95% confidence level and a ±5% margin of error. As our final sample exceeded this threshold, we ensured sufficient statistical power for reliable population-level inferences.

The protocol of this study was approved by the Scientific Council of Babeș-Bolyai University Cluj-Napoca under number 224/27.02.2024. For minor participants, written informed consent was obtained from their parents or legal guardians. Adult participants provided their own written informed consent. Participation was voluntary, with informed consent obtained, and all responses were anonymized to ensure data privacy and ethical compliance.

### 2.2. Data and Measurement

This study used a structured data collection methodology to evaluate adolescent BMI, dietary practices, PA levels, the risk of EDs, and diet culture knowledge. Data were gathered via a Google Forms survey, which participants accessed through a unique link provided during classroom sessions and completed using a mobile device. The demographic profile questions within the survey included inquiries about the participant’s high school, residence, age, gender, and prior consultations with a specialist for weight or mental health concerns.

#### 2.2.1. Anthropometric Measurements

Participants’ height and weight were self-reported. Height was reported in centimeters and converted to meters (m) for analysis. Weight was reported in kilograms. Body Mass Index (BMI) was calculated as weight (kg) divided by height (m) squared. To comprehensively categorize weight status, we employed a dual approach, utilizing both age- and gender-specific BMI percentiles and BMI z-scores (standard deviation scores, SDS), in accordance with the WHO Child Growth Standards for individuals aged 5–19 years [45]. BMI percentiles were categorized as underweight (≤5th percentile), healthy weight (>5th to <85th percentile), overweight (≥85th to <95th percentile), and obese (≥95th percentile). Simultaneously, BMI z-scores (SDS) were calculated and the following classifications, as defined by the WHO, were used: severe thinness (<−3 SDS), thinness (≥−3 SDS to <−2 SDS), healthy weight (≥−2 SDS to ≤+1 SDS), overweight (>+1 SDS to ≤+2 SDS), obesity (>+2 SDS to ≤+3 SDS), and severe obesity (>+3 SDS). For the participants aged 19–20 years, we continued to use the WHO Child Growth Standards z-scores (SDS) classifications, ensuring consistency in applying growth references across the entire adolescent age range. This dual classification approach, using both percentiles and z-scores, enhances clinical interpretability, facilitates comparison with international growth references, and ensures methodological rigor appropriate for epidemiological analysis.

#### 2.2.2. Dietary Assessment: Medi-Lite Score

Adherence to the MedDiet was assessed using a modified Mediterranean Diet Adherence Score (Medi-Lite score), excluding the alcohol component [46]. The original Medi-Lite score is a 9-item instrument. The present study used a modified 8-item version, excluding the alcohol component due to the age of the participants and the cultural context. This adapted instrument evaluated the frequency and portion size-adjusted consumption of eight food groups: vegetables, legumes, fruits, cereals/tubers, dairy, fish, meat and meat products, and olive oil. Participants indicated their usual consumption for each food group by selecting from three pre-defined categories, representing different portion frequencies. For vegetables, fruits, and cereals, these categories represented low, moderate, and high intake (scored as 0, 1, and 2, respectively). Legume and fish consumption categories were defined on a weekly basis. In contrast, meat and dairy product categories were scored in reverse (2, 1, and 0 for low, moderate, and high intake, respectively), reflecting MedDiet principles. Olive oil use was categorized as “Occasional”, “Frequent”, or “Regular” (scored as 0, 1, and 2, respectively). The modified Medi-Lite score was calculated by summing the scores for each food group, resulting in a possible score range of 0 to 16. Higher scores indicate greater adherence to the MedDiet. Participants were then categorized into Low, Moderate, or High adherence groups based on tertiles of this score distribution: low adherence (<5 points), moderate adherence (5–8 points), and high adherence (≥9 points).

#### 2.2.3. Consumption of UPF

Consumption of UPF and beverages was assessed using five variables derived from self-reported questionnaires. These variables captured the frequency of consumption of fast food, sweets, high-salt foods, energy drinks, and sugar-sweetened beverages (SSBs) [47]. Participants indicated their consumption frequency for each item using a multipoint scale (e.g., (0) “Never”, (1) “Occasionally (once last week)”, (2) “2 to 4 times last week”, (3) “5 to 6 times last week”, (4) “At least once a day”). To determine an overall UPF consumption level, a total score was calculated by summing the numerical values assigned to each frequency category for fast food, sweets, high-salt foods, energy drinks, and SSBs. For reporting purposes, these original ordinal response categories were dichotomized into “Infrequent” (≤2 times/week) and “Frequent” (>2 times/week) consumption categories based on established nutritional guidelines [48].

#### 2.2.4. Physical Activity Assessment

PA was assessed using a modified subset of items from the PA Questionnaire for Children (PAQ-C) and Adolescents (PAQ-A), a validated self-report instrument measuring general PA over the past 7 days [49,50]. Due to time constraints, select items representing four domains were employed: spare time activity, activity during physical education (PE) classes, recess activity, and walking to/from school. Participants reported activity levels using item-specific response options reflecting varying frequencies and intensities. For leisure-time activity, options ranged from “little/no physical effort” (recoded as 1) to “≥7 times/week” (recoded as 5). School PE responses ranged from “No PE” (1) to “Always” (5). Recess activity options ranged from “Sat down” (1) to “Ran/played hard most of the time” (5). Walking to/from school responses ranged from “None” (1) to “≥4 times/week” (5). After recoding to fit the mentioned scale (1–5), all original responses were used to calculate a modified PA score (range: 1–5), representing the mean of these items. Participants were then categorized into activity quartiles: “Lowest” (1.00–2.75), “Lower-Middle” (2.76–3.25), “Upper-Middle” (3.26–4.00), and “Highest” (4.01–5.00), based on the distribution of scores.

#### 2.2.5. Eating Attitudes Test-26 (EAT-26)

ED risk was assessed using the EAT-26, a widely used and validated 26-item self-report questionnaire [36,51,52]. Participants respond to each item on a six-point Likert scale: “Always”, “Usually”, “Often”, “Sometimes”, “Rarely”, and “Never”. Following standard scoring procedures, responses were coded as follows: “Always” = 3, “Usually” = 2, “Often” = 1, and “Sometimes”, “Rarely”, and “Never” = 0. Item 25 (“Enjoy eating new and rich foods”) was reverse scored. The total EAT-26 score is the sum of all 26 item scores (range: 0–78), with higher scores indicating greater ED risk. A score of 20 or above is conventionally used as a clinical cutoff, indicating a potential need for further evaluation. A dichotomous variable was created to represent this clinical cutoff. In addition to the total score, three established subscales were calculated [53]: Dieting, reflecting avoidance of fattening foods and preoccupation with body size (Items 1, 6, 7, 10, 11, 12, 14, 16, 17, 22, 23, 24, 26); Bulimia and Food Preoccupation, reflecting thoughts about food and behaviors associated with bulimia nervosa (Items 3, 4, 9, 18, 21, 25); and Oral Control, reflecting self-control related to food and perceived pressure from others to gain weight (Items 2, 5, 8, 13, 15, 19, 20).

### 2.3. Statistical Analysis

Data were analyzed using IBM Statistical Package for the Social Sciences (SPSS) Statistics for Mac OS (Version 29.0.1) and Integrated Development Environment for (R RStudio) Desktop for MacOS (Version 2023.06 “Mountain Hydrangea”). Descriptive statistics (means, standard deviations, frequencies, percentages) were used to summarize sociodemographic characteristics, BMI, dietary profile, PA levels, and EAT-26 scores.

Chi-square tests of independence were employed to examine associations between categorical variables, such as gender and weight status. Independent samples t-tests were conducted to compare mean differences in continuous variables (e.g., BMI) between groups (e.g., males vs. females).

To evaluate the psychometric properties of the EAT-26 within this sample, a confirmatory factor analysis (CFA) was performed using maximum likelihood estimation. The hypothesized three-factor structure (Dieting, Bulimia and Food Preoccupation, and Oral Control) was tested against the observed data. Model fit was assessed using the Comparative Fit Index (CFI), Tucker–Lewis Index (TLI), Root Mean Square Error of Approximation (RMSEA), and Standardized Root Mean Square Residual (SRMR). Given the suboptimal fit of the initial CFA model, an exploratory factor analysis (EFA) was subsequently conducted using principal axis factoring extraction with oblimin rotation in SPSS version 29.0. Items with communalities and factor loadings lower than 0.30 were considered inadequate and removed from the scale. Reliability analysis was performed to assess internal consistency using Cronbach’s alpha.

Finally, binary logistic regression was conducted to identify significant predictors of scoring at or above the clinical cutoff on the EAT-26. The model included gender, residence, age, BMI, Medi-Lite score, UPF score, PA score, and consultation with a specialist for weight or mental health as potential predictors. Model fit was assessed using the Hosmer–Lemeshow goodness-of-fit test. Statistical significance was set at *p* < 0.05 for all analyses.

## 3. Results

### 3.1. Demographic Characteristics and Anthropometric Measurements

The study included 423 adolescents (Table 1), with a slightly higher proportion of females (52.7%, *n* = 223) than males (47.3%, *n* = 200). The participants’ ages ranged from 13 to 20 years, with the largest proportion falling in the 15–16 years age category (58.9%, *n* = 249), followed by 17–18 years (22.5%, *n* = 95), 13–14 years (15.2%, *n* = 64), and 19–20 years (3.3%, *n* = 14). The mean age of participants was 16.0 years (SD = 1.3). Most adolescents resided in urban areas (72.3%, *n* = 306), while 27.7% (*n* = 117) resided in rural areas.

The mean BMI for the sample was 21.45 kg/m^2^ (SD = 3.80), with a median of 20.96 kg/m^2^. The BMI values ranged from 12.91 kg/m^2^ to 39.90 kg/m^2^. Based on BMI percentiles, the distribution of weight status among participants indicated that the majority were classified as having a healthy weight (77.4%, *n* = 318). However, a notable proportion were classified as overweight (16.3%, *n* = 67) or obese (5.6%, *n* = 23). A small percentage were categorized with thinness (0.5%, *n* = 2) or severe obesity (0.2%, *n* = 1). The BMI percentile distribution indicates that the majority of participants fell within the 50th to 75th percentiles, with 25.8% of adolescents at the 75th percentile. Lower percentiles (≤15th) accounted for 14.6% of the sample, while higher percentiles (≥95th) comprised 13.4%.

A significant association was observed between gender and weight status (χ^2^(4) = 10.219, *p* = 0.037). While the majority of both male and female adolescents were classified as having a healthy weight, a higher proportion of males presented with overweight, and obesity compared to females. Conversely, all cases of thinness were observed in females. Independent sample *t*-tests revealed gender-based differences in BMI. Females exhibited a significantly lower mean BMI (M = 20.95, SD = 3.99) than males (M = 22.01, SD = 3.48; t(416) = −2.881, *p* = 0.004), with a small to moderate effect size (Cohen’s *d* = −0.282). No significant gender difference was observed in age (*p* = 0.680).

Residence also played a significant role in BMI and age variations. Adolescents residing in rural areas had a significantly higher mean BMI (M = 22.68, SD = 4.71) compared to those in urban areas (M = 20.98, SD = 3.27; t(416) = 4.186, *p* < 0.001), with a moderate effect size (Cohen’s *d* = 0.457). Additionally, adolescents from rural areas were significantly older (M = 16.14, SD = 1.40) than their urban counterparts (M = 15.67, SD = 1.33; t(420) = 3.213, *p* = 0.001), with a small to moderate effect size (Cohen’s *d* = 0.350). Furthermore, a significant relationship was found between gender and residence (χ^2^(1) = 7.194, *p* = 0.007), with a higher proportion of females residing in rural areas than males.

### 3.2. Dietary Profile

The overall adherence to the MediDiet, as assessed by the Medi-Lite score, ranged from 2.00 to 13.00, with a mean of 5.74 (SD = 1.77) and a median of 6.00 (Table 2; Appendix A). The distribution of scores indicated that the majority of participants exhibited moderate adherence (69.5%, *n* = 294), followed by low adherence (24.6%, *n* = 104), high adherence (5.7%, *n* = 24), and very high adherence (0.2%, *n* = 1). Females had a higher mean Medi-Lite score (M = 5.96, SD = 1.82) than males (M = 5.51, SD = 1.67), with a statistically significant difference (t(421) = 2.662, *p* = 0.008) and a moderate effect size (Cohen’s *d* = 0.259). Urban residents had a higher mean dietary profile score (M = 5.84, SD = 1.71) compared to rural residents (M = 5.50, SD = 1.90), a statistically significant difference (t(421) = −1.797, *p* = 0.037, one-tailed; *p* = 0.073, two-tailed) with a small effect size (Cohen’s *d* = −0.195).

Consumption frequencies of select food groups were assessed (Table 2). Participants were dichotomized into infrequent or frequent consumers based on reported intake. Notably, a majority reported frequent consumption of fast food (61.5%), sweets (75.7%), and SSBs (62.2%). High-salt food consumption was slightly lower but still frequent in the majority (55.6%). In contrast, energy drink consumption was predominantly infrequent (81.1%). Chi-square analyses revealed a significant association between gender and the frequency of sweets consumption (χ^2^(1) = 13.680, *p* < 0.001). Females (57.8%) reported significantly higher rates of frequent sweets consumption than males (42.2%). No significant gender differences were observed for the other food and beverage categories. A significant association was observed between residence and frequent sweets consumption (χ^2^(1) = 5.776, *p* = 0.016). Urban residents reported a higher frequency of sweets consumption (69.4%) than rural residents (30.6%). Furthermore, a significant association was found between residence and frequent energy drink consumption (χ^2^(1) = 7.509, *p* = 0.006), with urban residents reporting a higher frequency of consumption (60.0%) compared to rural residents (40.0%). No significant associations were found between residence and frequent consumption of fast food (*p* = 0.174), high-salt foods (*p* = 0.382), or SSBs (*p* = 0.104).

Analyses of UPF and beverage consumption revealed several noteworthy patterns. Frequent fast-food consumers (>2 times/week) were slightly older (*p* = 0.039) and had slightly lower PA scores (*p* = 0.031) compared to infrequent consumers. Still, effect sizes were small (Cohen’s *d* = −0.207 and 0.216, respectively). No significant differences between frequent and infrequent fast-food consumers were found in BMI or overall dietary profile. When examining sweets consumption, infrequent consumers exhibited a slightly higher BMI (*p* = 0.021) than frequent consumers, although the effect size was moderate (Cohen’s *d* = 0.264). No other significant differences were found between frequent and infrequent sweets consumers. For energy drink consumption, frequent consumers were significantly older than infrequent consumers (*p* < 0.001), with a moderate effect size (Cohen’s *d* = −0.478). Still, no differences were found in BMI, dietary profile, or PA. Finally, no significant differences were observed between infrequent and frequent consumers of SSBs regarding age, BMI, dietary profile score, or PA score.

### 3.3. PA Levels

A modified PA score was computed based on responses to questions regarding spare time PA, activity during PE classes, activity during recess, and walking to/from school (Table 3). The score ranged from 1.00 to 5.00, with a mean of 3.33 (SD = 0.83) and a median of 3.25.

In terms of spare time activity, a substantial proportion of adolescents reported frequent engagement, with 31.7% (*n* = 134) reporting PA 5–6 times per week (“quite often”) and 30.0% (*n* = 127) reporting seven or more times per week (“very often”). During PE classes, a combined 53.6% of participants reported being “always” or “quite often” very active, though 13.2% (*n* = 56) reported being “hardly ever” active, and 6.1% (*n* = 26) indicated they did not participate in PE. Recess activity levels were generally moderate, with 28.1% (*n* = 119) reporting they “ran or played a little bit”, 27.7% (*n* = 117) reporting they “ran around and played quite a bit”, and 22.2% (*n* = 94) reporting they “ran and played hard most of the time”. Regarding active transport, 35.5% (*n* = 150) reported “none”, while the remaining participants reported frequencies ranging from once per week (17.7%, *n* = 75) to four or more times per week (16.3%, *n* = 69).

Males exhibited significantly higher PA scores (M = 3.51, SD = 0.85) compared to females (M = 3.17, SD = 0.78) (t(421) = −4.259, *p* < 0.001), with a large effect size (Cohen’s *d* = −0.415). Furthermore, urban residents exhibited significantly higher PA scores (M = 3.39, SD = 0.84) than rural residents (M = 3.18, SD = 0.79) (t(421) = −2.319, *p* = 0.010, one-tailed; *p* = 0.021, two-tailed), with a small effect size (Cohen’s *d* = −0.252).

### 3.4. Symptoms and Concerns Characteristic of EDs

Descriptive analysis of the EAT-26 revealed a mean total score of 15.75 (SD = 14.07, Range = 0–73), with a median of 11.00. The mean scores for the Dieting, Bulimia, and Oral Control subscales were 8.75 (SD = 8.35, Range = 0–38), 2.66 (SD = 3.72, Range = 0–18), and 4.35 (SD = 4.23, Range = 0–21), respectively. The frequency distributions for the total score and subscales indicated a positive skew, with most participants scoring on the lower end of the scales. Critically, 26.5% of participants (*n* = 112) presented with EAT-26 scores at or above the clinical cutoff of 20. Appendix A presents the item-level distribution of responses to the Eating Attitudes Test-26 (EAT-26).

A confirmatory factor analysis, employing maximum likelihood estimation, was conducted to examine the hypothesized three-factor structure of the EAT-26 within the present sample. However, the results indicated a suboptimal fit of the model to the observed data, suggesting that the proposed factor structure was not fully supported. Specifically, the fit indices did not meet the acceptable fit criteria, as per Hu and Bentler [54]. The CFI was found to be 0.781, the TLI was 0.759, the RMSEA was 0.079, and the SRMR was 0.087. While these values approach the thresholds, they do not meet the stringent criteria indicative of a good model fit. The internal consistency reliability of the EAT-26 was assessed using Cronbach’s alpha. The total scale demonstrated excellent reliability (α = 0.908), and the Dieting (α = 0.867), Bulimia and Food Preoccupation (α = 0.820), and Oral Control (α = 0.717) subscales showed good to acceptable internal consistency. These findings support the reliability of the EAT-26 in measuring ED risk among adolescents in this sample.

Nevertheless, a more granular analysis of individual item loadings presented a more expectant picture. Except for item EAT26, all factor loadings were statistically significant and ranged from 0.268 to 0.932, suggesting that these items generally aligned well with their intended latent factors (Table 4). This indicates a reasonable convergence of most items onto their respective factors. Moreover, the correlations between the three factors were statistically significant, as presented in Table 5, revealing a notable degree of interrelation.

Modifications were deemed necessary due to the poor model fit. Two major adjustments were made to improve the model. First, items with factor loadings less than 0.40 were removed. This resulted in the removal of two items: EAT26 (“Enjoy trying new rich foods”) and EAT19 (“Display self-control around food”). Second, three residual covariances were added based on modification indices and theoretical considerations: EAT11 with EAT14 (MI = 108.301), EAT11 with EAT12 (MI = 52.049), and EAT8 with EAT13 (MI = 54.397). These covariances reflect shared variance between items related to body image preoccupation and perceptions of others’ opinions. Also, based on modification indices and theoretical considerations, one cross-loading was added: EAT2 on “Dieting”. Following these changes, the model’s fit improved: CFI = 0.868, TLI = 0.843, RMSEA = 0.082 (90% CI = 0.067–0.076), and SRMR = 0.067. Cronbach’s alpha was used to assess the internal consistency of each factor in the modified three-factor model. With a value of 0.898, factor 1 (Dieting) demonstrated excellent internal consistency. Factor 2 (Bulimia) also demonstrated good internal consistency, with a value of 0.825. Factor 3’s (Oral Control) consistency was lower, with a value of 0.744, which is acceptable. However, it suggests that the items in this factor may need to be refined or reconsidered to improve their reliability.

Due to a suboptimal fit for the hypothesized three-factor model, an EFA was conducted, revealing a more suitable two-factor solution explaining 44.47% of the total variance (Table 6). Factor 1, labeled “Dieting, Body Image Concerns, and Food-Related Guilt”, included items related to dieting behaviors, thinness preoccupation, caloric awareness, and guilt around food consumption. Factor 2, labeled “Social Pressure and Loss-of-Control Behaviors”, captured items reflecting external social pressure to eat, binge-eating episodes, and compensatory behaviors such as vomiting. Communalities ranged from 0.071 to 0.840, with notably low values observed for items EAT26 (“Enjoy trying new rich foods”, extraction = 0.071) and EAT19 (“Display self-control around food”, extraction = 0.080). These two items were subsequently removed from the analysis. Their exclusion led to a significant improvement in the internal consistency of the scale, increasing Cronbach’s alpha from 0.908 (original 26-item scale) to 0.920 (final 24-item scale). The revised two-factor structure, thus, demonstrated robust psychometric validity suitable for subsequent analyses of eating disorder risk predictors in this adolescent population.

A binary logistic regression was conducted to examine the effects of gender, residence, age, BMI, dietary profile evaluation, PA, and knowledge about seeking help on the likelihood of participants scoring at or above the clinical threshold on the EAT-26 (Table 7). The overall logistic regression model was statistically significant, χ^2^(7) = 38.24, *p* < 0.001, accounting for 12.6% of the variance in EAT-26 clinical status (Nagelkerke R^2^) and correctly classifying 73.9% of cases. The Hosmer–Lemeshow goodness-of-fit test was non-significant, χ^2^(8) = 4.69, *p* = 0.791, indicating adequate model fit.

Results indicated that consulting a specialist for weight or mental health was a statistically significant predictor of scoring at or above the EAT-26 clinical threshold (*p* < 0.001), with an odds ratio (OR) of 3.76, 95% CI [2.20, 6.16], and an unstandardized odds ratio (uOR) of 3.92. This suggests that knowledge about when to seek professional help may be indicative of existing ED symptoms rather than a protective factor.

Gender also reached the conventional significance level (aOR = 1.61, *p* = 0.044); (uOR = 1.60, *p* = 0.034), females exhibited 1.60 times higher odds of scoring at or above the clinical threshold than males. Similarly, BMI approached significance (*p* = 0.075), with a uOR of 1.06 (*p* = 0.035), suggesting a potential trend whereby a higher BMI may be associated with increased odds of scoring at or above the EAT-26 clinical cutoff. Age (uOR = 1.08, *p* = 0.344), evaluation of dietary profile (uOR = 0.97, *p* = 0.673), UPF score (uOR = 0.92, *p* = 0.331), residence (uOR = 1.36, *p* = 0.195), and PA level (uOR = 0.87, *p* = 0.284) did not significantly predict the likelihood of scoring at or above the clinical threshold.

## 4. Discussion

### 4.1. Prevalence of Eating Disorder Risk Among Romanian Adolescents

According to this study, there is a considerable risk of EDs among Romanian adolescents, as 26.5% of participants scored higher than the EAT-26 clinical threshold. The findings of this study highlight BMI, gender, dietary patterns, PA levels, and prior specialist consultation as key factors associated with ED risk in Romanian adolescents. While BMI has historically been considered a significant risk factor, our results suggest that eating behaviors, psychological distress, and weight-related experiences may play a more critical role in shaping ED vulnerability. These findings support worldwide patterns while offering culturally particular information pertinent to preventative initiatives. EDs are a significant concern, with notable gender and population differences. In Finland, 17.9% of females and 2.4% of males have a lifetime prevalence of DSM-5 EDs, including anorexia, bulimia, and BED [55]. BED affects 1.32% of children and adolescents, with 3.0% showing subclinical symptoms [56]. In Spain, 30.1% of adolescents exhibit disordered eating behaviors [57].

The prevalence of EDs among adolescents in Romania is not directly addressed in the literature, but insights can be inferred from broader studies and local data. For example, BED affects approximately 1.32% of adolescents, with subclinical cases reaching 3.0% [56]. In this study, a descriptive analysis of the EAT-26 revealed a mean total score of 15.75 (SD = 14.07), with 26.5% of participants reaching the clinical threshold of 20 or above. This prevalence highlights the substantial risk of EDs in adolescents. While the distribution of scores was positively skewed, indicating that most participants exhibit subclinical levels, a significant minority demonstrate behaviors that warrant clinical attention. The subscale scores—dieting (M = 8.75), bulimia (M = 2.66), and oral control (M = 4.35)—indicate that dieting behavior is the most significant problem and is consistent with broader trends in adolescent populations, where body dissatisfaction and restrictive eating are common [58,59]. Lower mean scores for bulimia and oral control suggest that these behaviors may be less common but still present in a subset of participants.

### 4.2. Factor Structure and Psychometric Considerations of EAT-26

Confirmatory factor analysis CFA of the hypothesized three-factor structure revealed a suboptimal model fit, with CFI (0.781) and TLI (0.759) falling below recommended thresholds. The model adjustments led to a modest improvement in fit indices (CFI = 0.868, TLI = 0.843) by incorporating error covariances and removing low-performing items (EAT26 and EAT19). Despite these improvements, the model remained insufficiently robust for this adolescent sample, underscoring the potential need for further adaptation of the EAT-26 for younger populations. EAT-26 may not adequately capture the multidimensional nature of disordered eating in adolescents. Emerging evidence suggests that adolescents with disordered eating often exhibit comorbid emotional and behavioral difficulties, which may not be fully reflected in the EAT-26 [60].

The psychometric variability of the EAT-26 across adolescent populations suggests cultural influences on ED symptom presentation. Studies indicate that the original three-factor structure often fails to replicate, as seen in Israeli adolescents with differing factor structures among Jews, Muslims, and Christians [61] and in Irish adolescents where a six-factor EAT-18 model showed better fit [37]. Socio-cultural factors further shape ED risk, with urbanization and Westernized body ideals influencing symptomatology in Jordanian [62] and South African adolescents [63]. Differences in body dissatisfaction and risk behaviors between Spanish and Mexican adolescents further highlight the impact of cultural norms [64]. Given our study’s suboptimal fit, these findings suggest that cultural adaptation of the EAT-26 or alternative models like the EAT-18 may enhance its validity across diverse adolescent populations.

Internal consistency reliability was high for the overall scale (α = 0.908) and acceptable to excellent for the subscales: Dieting (α = 0.867), Bulimia (α = 0.820), and Oral Control (α = 0.717). These results support the scale’s reliability but indicate areas where improvement is needed, particularly for the Oral Control subscale. A detailed analysis of the factor loadings revealed that all items except EAT26 agreed well with their respective latent factors (factor loadings ranging from 0.268 to 0.932). The correlations between the factors—diet and bulimia (r = 0.269), diet and oral control (r = 0.134), and bulimia and oral control (r = 0.134)—were statistically significant. They suggested interconnected dimensions of ED behavior.

### 4.3. Gender Differences

Female participants were 1.6 times more likely than male participants to score at or above the clinical threshold on the EAT-26, although this finding narrowly missed statistical significance (*p* = 0.055). These results align with existing literature, which consistently reports a higher prevalence of EDs among women, with estimates indicating rates of approximately 3.2% in women compared to 1.3% in men and a gender ratio of about 2.5:1 favoring females [65]. This disparity may be attributed to sociocultural pressures, as adolescent girls experience heightened exposure to media-driven body ideals and appearance-related social comparison, which reinforce restrictive eating behaviors [66,67]. Body image dissatisfaction is a key mediating factor, as dissatisfaction with weight and shape strongly predicts ED onset and related health-risk behaviors [16,68]. Hormonal influences may further contribute, with early pubertal development and conditions such as PCOS exacerbating weight concerns and disordered eating tendencies [69].

### 4.4. Risk Factor Dynamics for Eating Disorders Risk

Higher BMI was a significant predictor (uOR = 1.06, *p* = 0.035), suggesting that weight status, likely influenced by body dissatisfaction and societal pressure, contributes to ED risk [16,70]. In multiple logistic regression, BMI approached statistical significance (*p* = 0.075), and its predictive power was weaker than anticipated. This finding contrasts with prior research, suggesting a direct link between elevated BMI and ED risk [15]. While BMI has been historically regarded as a major ED risk factor, our findings suggest that its role may be mediated by other behavioral and psychological factors. Body dissatisfaction, weight stigma, and dietary behaviors could be more influential in shaping ED vulnerability than BMI alone [71].

In this context, dietary habits play a crucial role, particularly regarding the consumption of UPF, which has been linked to mental health issues in children and adolescents. Likewise, adherence to the MedDiet in children has been associated with a lower risk of overweight and obesity [22,72,73,74]. Participants with lower dietary diversity (e.g., low consumption of fruits, vegetables, and legumes) and poor adherence to the MedDiet were not directly associated with EAT-26 scores. Still, they could be a protective factor for mental health in child and adolescent populations [73]. While adherence to the MedDiet is often linked to improved mental health and, potentially, reduced ED risk [75], our study did not assess the motivations behind dietary choices. Restrictive eating, emotional eating, or dieting behaviors, even within the context of a seemingly “healthy” dietary pattern like the MedDiet, could be indicative of disordered eating. Emerging evidence highlights the protective role of dietary patterns in mental health, particularly the MediDiet, which has been linked to lower risks of depression and anxiety [76]. This effect is attributed to anti-inflammatory and neuroprotective components, including omega-3 fatty acids, polyphenols, and folate, which modulate neurotransmitter activity, gut microbiota composition, and inflammatory pathways [77]. Adolescents adhering to a Western dietary pattern, high in processed foods, sugars, and saturated fats, are at increased risk of psychological distress, emotional dysregulation, and mood disorders [75]. The present study did not find a direct association between MD adherence and ED risk, but considering the bidirectional relationship between mental health and disordered eating, future research should examine whether dietary interventions targeting mental health outcomes can serve as a complementary strategy for ED prevention [78]. Public health initiatives should prioritize nutritional education and diet-based interventions as part of a holistic approach to adolescent mental health and ED prevention. For future studies to fully comprehend the intricate relationship between BMI and ED risk in adolescents, multi-dimensional assessments such as self-esteem, body image perception, and dietary habits should be included.

The null association between ultra-processed food (UPF) consumption and ED risk (uOR = 0.92, *p* = 0.331; aOR = 0.89, *p* = 0.172) is unexpected, and contrasts sharply with mounting evidence linking UPF intake to adverse mental health outcomes, specifically eating disorders [22,23,72]. This discrepancy demands a critical reevaluation of potential contributing factors. The cross-sectional design, while valuable for generating hypotheses, precludes causal inference. Reverse causality remains a plausible alternative: pre-existing ED symptoms, particularly those involving restrictive eating or fear of weight gain, might lead to a conscious or unconscious reduction in reported UPF intake. This could mask a potential etiological role of UPFs in ED development. Finally, unmeasured and residual confounding remains a critical concern. Factors such as underlying psychological distress, specific eating patterns (e.g., binge/emotional eating, which have been directly linked to UPF consumption [72], and socioeconomic disparities affecting food access likely exert independent and interactive influences on both UPF consumption and ED risk. These confounders could obscure a genuine relationship. It is also crucial to acknowledge that UPFs may differentially impact specific ED subtypes. Figueiredo et al. (2022) found significant associations between UPF intake and bulimic, binge eating, and other specified eating disorders, but not restrictive disorders, in a large French cohort [22]. Our study, using the EAT-26, may not have adequately captured these nuanced subtype-specific associations.

Participants who consulted specialists for weight or mental health issues were significantly more likely to be above the threshold (OR = 3.73, *p* < 0.001), reflecting existing symptoms rather than preventative behavior. Research indicates that mental health disorders, including anxiety and depression, significantly correlate with obesity in adolescents, suggesting that those with weight issues may already be experiencing psychological distress [79,80]. Other factors, including age, location, diet adherence, and PA, were not significant predictors [18], highlighting the complex, multifactorial nature of EDs [81].

Our findings indicate that PA levels did not significantly predict ED risk, which diverges from research linking compulsive exercise to disordered eating [82]. One possible explanation is that our PA measure primarily captured general activity levels rather than compulsive or excessive exercise behaviors, which are more closely associated with ED pathology [83]. Prior studies suggest that exercise motives, intensity, and compulsivity—rather than PA volume alone—are stronger predictors of ED risk [84,85]. Additionally, orthorexia nervosa, characterized by an excessive focus on “clean eating” and exercise, has been linked to disordered eating behaviors among young adults [83]. The increasing sedentarization of adolescents and the decline in structured PA participation further complicate this relationship, as lower perceived fitness levels have been associated with body dissatisfaction and restrictive eating behaviors [85]. Future studies should incorporate validated compulsive exercise scales and assess exercise motives and PA-related psychological traits to refine our understanding of the PA-ED relationship.

However, the explanatory power of the logistic regression model (Nagelkerke R^2^ = 12.6%) suggests that additional variables may influence disordered eating behaviors beyond those captured in the current study. In particular, mental health variables (e.g., depression, anxiety), social media exposure, and parental influence may function as unmeasured confounders that partially explain the variance in EAT-26 outcomes.

While this study primarily examined BMI, dietary habits, and PA, it is important to acknowledge the broader sociocultural factors contributing to ED risk, such as diet culture, social media exposure, and weight stigma [86]. Although we did not directly measure these influences, existing research suggests that unrealistic body standards and weight-related criticism can shape how adolescents perceive their bodies and approach eating [30]. Social media platforms often promote unrealistic body ideals, contributing to body dissatisfaction and disordered eating behaviors in adolescents and young adults [87,88,89,90]. The content and engagement patterns associated with platforms such as Instagram or YouTube can exacerbate unhealthy behaviors, including restrictive eating, binge eating, and purging [91]. Parental influence, particularly in the context of authoritarian parenting styles, may also heighten ED risk. This relationship can be mediated by psychological factors, such as depression and social anxiety, which underscores the multifactorial nature of eating disorder development [92]. Given that dieting behaviors were the most common concern in this study, media-driven body ideals and societal pressure to lose weight may have influenced participants’ attitudes toward food and body image. Additionally, weight stigma—especially for individuals with higher BMIs—has been linked to increased emotional distress and unhealthy eating behaviors, underscoring the need for prevention strategies that address both the physical and psychological aspects of ED risk [93].

The need for focused, gender-sensitive interventions is highlighted by the high prevalence of ED symptoms and gender inequalities, especially among male adolescents. These initiatives can potentially have a significant impact because they prevent EDs, encourage healthy weight management, and enhance mental health support for early detection [94,95]. The shortcomings of the EAT-26 in teenagers underscore the need for improved, developmentally appropriate tools and long-term studies, which may lead to better comprehension and more practical solutions to these problems.

### 4.5. Physical Activity Levels

The results of the current study showed that teenagers’ mean score on the modified PA scale was 3.33 (SD = 0.83), indicating moderate overall activity levels. While a significant proportion reported frequent engagement in leisure-time exercise—with 61.7% exercising at least five times per week—participation in structured PA settings, such as PE classes and active transport, was lower. These findings align with previous research, which highlights that the WHO-recommended PA guidelines—requiring at least 60 min of moderate-to-vigorous physical activity (MVPA) daily and at least three days of muscle-strengthening activities weekly—are unmet by more than 20% of adolescents globally. Furthermore, the current study supports evidence of gender disparities in PA levels, with boys (23%) more likely than girls (17%) to meet these guidelines [96].

The results of this study revealed mixed levels of PE classes, with 53.6% of participants reporting being active “always” or “fairly often”. In comparison, 19% stated they were active infrequently or never. These findings raise concerns about the accessibility and effectiveness of PE programs in schools. Existing research indicates that participation in PE classes is positively associated with overall PA levels among adolescents across multiple countries [97,98]. Factors such as lesson context, teacher behavior, and class location significantly influence participation, with outdoor lessons and higher motor content shown to increase MVPA [99]. Although obstacles like poor body image and low perceived competence prevent people from participating, exercise improves mental health and lowers the risk of EDs [100,101]. Studies have shown that adopting inclusive practices—such as reducing body-related remarks, promoting self-compassion, and emphasizing enjoyment over appearance—is essential to ensuring that all students benefit from PA.

Most participants exhibited low-intensity or sedentary behavior; only 22% reported engaging in vigorous activity during breaks. This finding aligns with studies demonstrating a significant increase in sedentary behavior during adolescence, with device-measured sedentary time rising by 27.9 to 140.7 min per day over 1–4 years [102] and more than 60% of sedentary time occurring at school, on the weekends, and after school [103]. Structured interventions are essential to address rising adolescent inactivity, maximize health benefits, and encourage more strenuous PA during breaks. Active commuting to school (ACS) among adolescents has decreased in recent years, with its prevalence differing across countries and regions [104,105]. Factors such as school distance and parental concerns about traffic significantly affect ACS rates [106]. Notably, 35% of participants indicated they do not actively commute, highlighting critical structural challenges, including unsafe road conditions, inadequate urban planning, and a societal dependence on motorized transportation.

### 4.6. Dietary Habits, MedDiet, and Ultra-Processed Food Consumption

The MedDiet is widely recognized for its health benefits and sustainability, emphasizing fruits, vegetables, whole grains, legumes, nuts, and olive oil while limiting red meat and processed foods [25].

The Medi-Lite score, a validated tool for assessing adherence to the MedDiet, evaluates the consumption of nine key food groups: fruits, vegetables, cereals, legumes, fish, meat, dairy, alcohol, and olive oil. Scores range from 0 to 18, with higher values reflecting stronger adherence to MedDiet principles [107]. In this study, the Medi-Lite score revealed that 69.5% of participants demonstrated moderate adherence to the MedDiet, while 24.6% were categorized as having low adherence. Significant deficiencies were observed in vegetables (36.2% consuming < 1 portion/day), legumes (57% consuming < 1 portion/week), and fruit consumption (40.2% consuming < 1 portion/day), indicating gaps in dietary diversity that are essential for the development and health of adolescents.

Excessive meat consumption (50.1% consuming > 1.5 portions/day) and low cereals, dairy, and fish intake contribute to dietary imbalances that may promote restrictive eating behaviors and body dissatisfaction. While 98% of males and 90% of females exceed the recommended red meat intake, fish consumption remains critically low, with 48.7% eating less than one portion per week and 7.6% of males and 10.9% of females abstaining entirely. In females, these patterns lead to excessive protein intake, increasing the risk of micronutrient deficiencies, particularly calcium, vitamin D, and iron [108]. Given that 26.5% of adolescents in this study scored above the clinical threshold for EDs, dietary education should emphasize balance over restriction to support physical and mental health. 

Participants’ dietary habits deviate from global health recommendations, with 43% consuming less than one portion of cereals daily—well below the WHO’s 5–9 servings—raising concerns about fiber deficiency and digestive health. Similarly, 42.1% consume insufficient dairy, increasing the risk of calcium and vitamin D deficiencies that may compromise bone health. Olive oil, a key source of healthy fats, is regularly consumed by only 27.4%, limiting cardiovascular benefits. Hydration is also inadequate, with only 25.1% drinking more than 1.5 L of water daily, while 17.7% fall below WHO hydration guidelines, potentially impacting overall well-being [109].

The results of this study indicate a high prevalence of UPF consumption among adolescents, with 61.5% frequently consuming fast food, 75.7% consuming sweets, and 62.2% consuming SSBs. These findings align with previous research linking high UPF intake to an increased risk of obesity, disordered eating behaviors, and negative mental health outcomes [22,23]. Gender differences were also evident, with females reporting significantly higher consumption of sweets than males (57.8% vs. 42.2%, *p* < 0.001). This result is consistent with studies indicating that emotional eating, often triggered by stress and negative effect, is a key factor in high UPF consumption among adolescents [110]. Additionally, urban adolescents reported more frequent consumption of sweets and energy drinks than their rural counterparts (*p* = 0.016, *p* = 0.006, respectively), which may reflect greater accessibility and increased exposure to unhealthy food environments in urban areas [111]. Interestingly, this study found that frequent sweet consumers had a slightly lower BMI than infrequent consumers (*p* = 0.021), and it may suggest a cycle of compensatory dietary restriction, a pattern previously observed among adolescents with tendencies toward disordered eating behaviors [22]. Furthermore, frequent fast-food consumption was associated with PA scores (*p* = 0.031), supporting evidence that high UPF intake is linked to sedentary lifestyles and an increased risk of weight-related issues [111].

Beyond BMI-focused interventions, the results highlight the necessity of prevention strategies. Nutrition education should prioritize balance over restriction and discourage unhealthy dieting trends, as dieting behaviors are the most common ED concern. Furthermore, fostering a positive relationship with PA is crucial, especially for teenagers participating in competitive sports, as the dangers of excessive exercise are underscored. Finally, since UPF use is linked to an increased risk of EDs, policies that increase teenagers’ access to complete, nutrient-dense foods ought to be put first. Given that male and female adolescents exhibit ED risk in different ways, gender-sensitive approaches should be incorporated into future public health initiatives.

## 5. Limitations

Several limitations should be considered when interpreting our findings. First, this study’s cross-sectional design precludes any inference of causality, limiting our ability to determine temporal relationships between potential risk factors and disordered eating behaviors. Future longitudinal studies are needed to track dietary behaviors, PA patterns, and psychological factors over time to better understand their role in ED onset and progression. Second, the reliance on self-reported measures—including anthropometric data, dietary intake, and PA—introduces the possibility of recall and reporting biases. Specifically, self-reported height and weight may lead to underestimation of BMI, particularly among overweight adolescents, thereby underestimating overweight prevalence [112]. This is a significant concern, as inaccurate BMI values could influence the associations observed with other variables. Similarly, self-reported dietary intake is prone to underreporting, especially of energy-dense, nutrient-poor foods, influenced by factors like body dissatisfaction and social approval needs [113]. Moreover, self-reported PA levels may be overestimated due to social desirability, with interventions potentially exacerbating this bias [114]. The combined effect of these self-report biases across multiple key variables could significantly impact the observed relationships and may partially explain the lack of significant associations between dietary patterns, PA, and ED risk. Third, we used a convenience sample of adolescents from seven high schools in Cluj-Napoca, which may restrict the generalizability of results to other Romanian regions or age groups. The voluntary nature of participation may introduce selection bias, as adolescents with eating concerns or body image issues might be more or less likely to participate, skewing the prevalence estimates of ED risk. Additionally, the distribution of participants between urban and rural areas was considered, but limited sample sizes within each area might restrict the ability to detect significant regional differences in ED risk factors. Though residence was included as a covariate in the regression models, substantial regional variations in lifestyle, resource access, and cultural norms related to food and body image may not be fully captured by our study. Fourth, while the EAT-26 is a widely recognized screening tool, it can overestimate ED risk and does not fully capture subclinical or partial-syndrome EDs. Fifth, we did not account for important sociocultural determinants—such as social media exposure, family influence, and weight stigma—which may substantially affect adolescents’ eating behaviors and body image perceptions. This study did not include socioeconomic status (SES) indicators, which may influence dietary habits, physical activity levels, and ED risk. SES disparities can affect food accessibility, healthcare utilization, and exposure to stressors related to disordered eating. Future research should integrate SES measures to provide a more comprehensive understanding of socioeconomic influences on ED susceptibility. Sixth, another potential limitation of this study is the inclusion of a small number of participants aged 19 years and older, which introduces heterogeneity in BMI comparisons because adult BMI references differ from adolescent growth standards. However, a sensitivity analysis excluding these adult participants showed no substantial changes in the main outcomes, suggesting that their inclusion did not materially bias our findings. Finally, although we examined BMI as a potential risk factor, future research incorporating more nuanced assessments (e.g., body image dissatisfaction, diet culture awareness, mental health symptoms) might yield a deeper understanding of how these interrelated factors influence ED susceptibility. Future longitudinal and mixed-methods studies are needed to address these limitations, enabling clearer insights into the complexities of ED risk and fostering more precise, context-specific prevention and intervention strategies.

## 6. Conclusions

This cross-sectional study highlights a substantial burden of ED risk among Romanian adolescents, with 26.5% surpassing the EAT-26 clinical cutoff. Although a higher BMI was moderately associated with increased ED vulnerability, its predictive value diminished when considered alongside other variables. In contrast, female gender and prior consultation with a specialist for weight or mental health concerns consistently emerged as stronger predictors of ED risk. These findings suggest that the interplay between psychosocial factors, gender-specific pressures, and existing psychological distress may more profoundly shape ED onset than anthropometric measures alone.

Notably, neither MedDiet adherence nor PA levels independently predicted disordered eating in this sample. This result underscores the multifaceted nature of ED risk and points to the potential role of broader sociocultural determinants—such as stigma, diet culture, and media influences—that were not explicitly assessed. While the utility of the EAT-26 as a screening instrument is supported by its high internal consistency, its limitations in fully capturing subclinical symptoms and cultural expressions of disordered eating necessitate cautious interpretation.

From a public health perspective, the results reinforce the need for comprehensive, culturally tailored strategies for early detection, prevention, and intervention. Future research should prioritize longitudinal studies to track ED risk factors over time, establishing causal pathways. Qualitative research is essential to explore adolescents’ lived experiences, providing insights into sociocultural factors, body image, and eating behaviors. Additionally, intervention studies are needed to evaluate targeted interventions in various settings (e.g., schools, families, communities) and diverse populations. This research must also refine assessment tools, creating sensitive and culturally appropriate instruments for identifying ED risk in adolescents.

Addressing this complex issue requires a multi-pronged approach involving concerted action from schools, families, and policymakers. Schools play a crucial role in implementing comprehensive health education programs that promote body positivity, media literacy, and healthy eating habits, without inadvertently promoting dieting or weight-centric approaches. Schools should also provide readily accessible early identification and support services for students at risk. Families can contribute by fostering open communication about body image and eating, modeling healthy behaviors, and seeking professional help when needed, avoiding stigmatizing language or weight-focused discussions. Policymakers have a responsibility to address systemic factors, such as regulating social media content that promotes unrealistic body ideals and promoting policies that ensure equitable access to mental health care and nutritious food options for all adolescents, regardless of socioeconomic status. A coordinated effort across these sectors is paramount to effectively mitigate the growing burden of eating disorders among young people.

## Figures and Tables

**Table 1 nutrients-17-01067-t001:** Sociodemographic and weight-related characteristics of participants (*N* = 423).

Characteristic	Category	Frequency (*n*)	Percent (%)
Gender	Female	223	52.7
	Male	200	47.3
Age (years)	13–14	64	15.2
	15–16	249	59
	17–18	95	22.5
	19–20	14	3.3
Mean Age (years) (SD)	Mean (SD)	15.80 (1.36)	
Residence	Rural	117	27.7
	Urban	306	72.3
BMI (kg/m^2^)	Mean (SD)	21.45 (3.80)	
	Median	20.96	
	Range	12.91–39.90	
BMI Percentile Distribution	5th	24	5.8
	10th	19	4.6
	15th	17	4.1
	25th	41	10.0
	50th	69	16.8
	75th	106	25.8
	85th	48	11.7
	90th	19	4.6
	95th	28	6.8
	97th	13	3.2
	99th	14	3.4
	100th	13	3.2
Weight Status	Thinness	2	0.5
	Healthy weight	318	77.4
	Overweight	67	16.3
	Obesity	23	5.6
	Severe obesity	1	0.2

Note: BMI = Body Mass Index; SD = Standard Deviation. Percentages reflect valid values and may not add up to exactly 100% due to rounding or missing data.

**Table 2 nutrients-17-01067-t002:** Dietary profile of participants.

Variable	Category	Frequency (*n*)	Percent (%)
Medi-Lite Score	Low	104	24.6
	Moderate	294	69.5
	High	25	5.9
Medi-Lite Score (Mean ± SD, Range)		5.74 ± 1.77	2.00–13.00
UPF Score (Mean ± SD, Range)		8.15 ± 3.20	0.00–19.00
Frequency of eating fast foods	Infrequent	163	38.5
	Frequent	260	61.5
Frequency of consuming sweets	Infrequent	103	24.3
	Frequent	320	75.7
Frequency of consuming high-salt foods	Infrequent	188	44.4
	Frequent	235	55.6
Frequency of consuming energy drinks	Infrequent	343	81.1
	Frequent	80	18.9
Frequency of consuming SSBs	Infrequent	160	37.8
	Frequent	263	62.2

Note: BMI = Body Mass Index; SD = Standard Deviation; SSBs = sugar-sweetened beverages; UPF = ultra-processed food. Percentages reflect valid values and may not add up to exactly 100% due to rounding or missing data.

**Table 3 nutrients-17-01067-t003:** PA levels of participants.

Variable	Category	Frequency (*n*)	Percent (%)
PA Score Quartiles	Low Score (1.00–2.75)	130	30.7
	Lower Middle Score (2.76–3.25)	83	19.6
	Upper Middle Score (3.26–4.00)	136	32.2
	High Score (4.01–5.00)	74	17.5
Spare Time PA	All or most of my free time was spent doing things that involve little physical effort.	14	3.3
	I sometimes (1–2 times/week) did physical things in my free time.	47	11.1
	I often (3–4 times/week) did physical things in my free time.	101	23.9
	I quite often (5–6 times/week) did physical things in my free time.	134	31.7
	I very often (7 or more times/week) did physical things in my free time.	127	30
PA During PE Classes	I don’t do PE	26	6.1
	Hardly ever	56	13.2
	Sometimes	114	27
	Quite often	108	25.5
	Always	119	28.1
PA During Recess	Sat down (talking, reading, doing schoolwork)	13	3.1
	Stood around or walked around	80	18.9
	Ran or played a little bit	119	28.1
	Ran around and played quite a bit	117	27.7
	Ran and played hard most of the time	94	22.2
Walking Before or After School	None	150	35.5
	Occasionally (1 time/week)	75	17.7
	2 times/week	80	18.9
	3 times/week	49	11.6
	4 times or more/week	69	16.3

Note: PE = Physical Exercise; SD = Standard Deviation. Percentages reflect valid values and may not add up to exactly 100% due to rounding or missing data.

**Table 4 nutrients-17-01067-t004:** Standardized coefficients and associated data for the EAT-26 Score.

Factor	Indicator	Estimate	SE	z-Value	*p*	95% CI Lower	95% CI Upper	R^2^
Dieting	EAT1	0.463	0.045	10.280	<0.001	0.374	0.551	0.214
Dieting	EAT6	0.440	0.047	9.400	<0.001	0.348	0.531	0.193
Dieting	EAT7	0.617	0.056	11.040	<0.001	0.508	0.727	0.381
Dieting	EAT10	0.721	0.043	16.830	<0.001	0.637	0.805	0.520
Dieting	EAT11	0.693	0.045	15.260	<0.001	0.604	0.782	0.480
Dieting	EAT12	0.691	0.044	15.640	<0.001	0.605	0.778	0.478
Dieting	EAT14	0.731	0.041	17.670	<0.001	0.650	0.812	0.535
Dieting	EAT16	0.691	0.051	13.540	<0.001	0.591	0.791	0.477
Dieting	EAT17	0.631	0.058	10.820	<0.001	0.516	0.745	0.398
Dieting	EAT22	0.753	0.033	22.520	<0.001	0.687	0.818	0.566
Dieting	EAT23	0.609	0.050	12.230	<0.001	0.512	0.707	0.371
Dieting	EAT24	0.677	0.044	15.470	<0.001	0.592	0.763	0.459
Dieting	EAT26	−0.062	0.057	−1.090	0.274	−0.173	0.049	0.004
Bulimia	EAT3	0.441	0.045	9.830	<0.001	0.353	0.529	0.194
Bulimia	EAT4	0.592	0.050	11.870	<0.001	0.494	0.689	0.350
Bulimia	EAT9	0.545	0.065	8.370	<0.001	0.417	0.672	0.297
Bulimia	EAT18	0.714	0.042	17.080	<0.001	0.632	0.795	0.509
Bulimia	EAT21	0.762	0.032	23.870	<0.001	0.700	0.825	0.581
Bulimia	EAT25	0.932	0.014	69.220	<0.001	0.906	0.959	0.869
Oral Control	EAT2	0.560	0.068	8.210	<0.001	0.426	0.694	0.314
Oral Control	EAT5	0.540	0.056	9.650	<0.001	0.430	0.649	0.291
Oral Control	EAT8	0.521	0.061	8.590	<0.001	0.402	0.640	0.271
Oral Control	EAT15	0.604	0.045	13.480	<0.001	0.516	0.692	0.365
Oral Control	EAT19	0.268	0.057	4.740	<0.001	0.157	0.380	0.072
Oral Control	EAT20	0.600	0.061	9.880	<0.001	0.481	0.719	0.360

Note: EAT = Eating Attitudes Test; CI = Confidence Interval. Factor loadings represent the correlation between each item and the underlying factor. *p* < 0.05 is considered statistically significant. R^2^ values represent the proportion of variance in each item explained by the factor.

**Table 5 nutrients-17-01067-t005:** Correlations among ED symptom subscales.

Correlation	Estimate	SE	z-Value	*p*-Value	Lower CI	Upper CI
Dieting and Bulimia	0.269	0.044	6.092	< 0.001	0.182	0.355
Dieting and Oral Control	0.134	0.048	2.767	0.006	0.039	0.229
Bulimia and Oral Control	0.134	0.044	3.059	0.002	0.048	0.221

Note: CI = Confidence Interval; SE = Standard Error. The table presents Pearson correlation coefficients.

**Table 6 nutrients-17-01067-t006:** Final two-factor structure of the EAT-26 with factor loadings.

Item	Item Description	Factor 1 Loadings Dieting, Body Image Concerns, and Food-Related Guilt	Factor 2 Loadings Social Pressure and Loss-of-Control Behaviors
1	Terrified about being overweight	0.584	
2	Avoid eating when hungry	0.593	
3	Preoccupied with food	0.387	
4	Have gone on eating binges	0.474	
6	Aware of calorie content	0.479	
7	Avoid high carbohydrate food	0.523	
10	Feel extremely guilty after eating	0.724	
11	Desire to be thinner	0.805	
12	Think about burning calories	0.781	
14	Thought of having fat on body	0.816	
16	Avoid foods with sugar	0.585	
17	Eat diet foods	0.503	
18	Food controls life	0.553	
21	Too much time and thought to food	0.699	
22	Uncomfortable after eating sweets	0.734	
23	Engage in dieting behavior	0.546	
24	Like stomach empty	0.655	
25	Impulse to vomit after meals	0.786	
5	Cut food into small pieces		0.225
8	Others prefer if ate more		0.568
9	Vomit after eating		0.435
13	Others think I’m too thin		0.680
15	Take longer to eat meals		0.403
20	Others pressure me to eat		0.642

Note: Items EAT19 and EAT26 were excluded due to low communalities and insufficient factor loadings. Factor loadings below 0.30 are not displayed.

**Table 7 nutrients-17-01067-t007:** Simple and multiple binary logistic regression predicting the likelihood of scoring at or above EAT-26 clinical cutoff (*N* = 418).

Predictor	uOR (Simple)	95% CI (Simple)	*p*-Value (Simple)	aOR (Multiple)	95% CI (Multiple)	*p*-Value (Multiple)
Gender (Female vs. Male)	1.6	[1.04, 2.46]	0.034 *	1.61	[1.01, 2.65]	0.044 *
Residence (Urban vs. Rural)	1.36	[0.86, 2.16]	0.195	1.06	[0.62, 1.79]	0.84
Age (years)	1.08	[0.92, 1.26]	0.344	1.07	[0.90, 1.28]	0.45
BMI	1.06	[1.00, 1.12]	0.035 *	1.05	[0.99, 1.12]	0.097
Medi Lite Score	0.97	[0.86, 1.10]	0.673	0.92	[0.80, 1.05]	0.198
Ultra Processed Food Consumption Score	0.92	[0.79, 1.08]	0.331	0.89	[0.75, 1.05]	0.172
Physical Activity Score	0.87	[0.67, 1.12]	0.284	0.98	[0.74, 1.31]	0.91
Consulting a specialist for weight or mental health (Yes vs. No)	3.92	[2.45, 6.29]	<0.001 ***	3.76	[2.20, 6.16]	<0.001 ***

Note: This table presents unstandardized odds ratios (uOR) from simple logistic regressions and adjusted odds ratios (aOR) from multiple logistic regressions. * *p* < 0.05, *** *p* < 0.001; CI, Confidence Interval.

## Data Availability

The data supporting this study’s findings are available from Babeș-Bolyai University. Access to the data can be requested from the corresponding author upon reasonable request with the permission of Babeș-Bolyai University. They are not publicly available due to privacy and ethical restrictions.

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
