# Peer review of "Eating Disorder Risk Among Adolescents: The Influence of Dietary Patterns, Physical Activity, and BMI"

_nutrients, 2025, doi:10.3390/nu17061067_

Round 1

Reviewer 1 Report

Comments and Suggestions for Authors

Overall Assessment

This study provides a valuable contribution to the understanding of eating disorder (ED) risks among adolescents in Romania. It employs a comprehensive approach by analyzing the interplay of demographic factors, BMI, dietary patterns, and physical activity (PA) in predicting ED risk. The use of the Eating Attitudes Test-26 (EAT-26) alongside dietary and physical activity assessments strengthens the study's methodological rigor. The findings are relevant for public health interventions aimed at improving adolescent health behaviors.

While the study is well-structured and methodologically sound, there are areas that need further refinement, particularly in data interpretation, statistical validation, and discussion of limitations. Below, I outline the strengths of the manuscript and areas that require improvement.

Strengths of the Study

  1. Timely and Relevant Topic
    • The rising prevalence of EDs among adolescents makes this study significant.
    • The study contributes context-specific evidence for Romania, which is underrepresented in ED research.
  2. Comprehensive Data Collection and Analysis
    • The study includes a large and diverse sample of 423 adolescents.
    • The use of validated tools (EAT-26, Medi-Lite, PAQ-C) ensures robust measurement of ED risk, dietary habits, and physical activity.
    • Statistical analysis is well-executed, including confirmatory factor analysis (CFA) for EAT-26 validation.
  3. Clear Identification of Risk Factors
    • Findings confirm that gender, BMI, and prior consultations with specialists are significant predictors of ED risk.
    • The study highlights the impact of UPF consumption and dietary adherence on ED risk, aligning with international research.
  4. Methodological Rigor
    • The study employs logistic regression to examine ED risk predictors.
    • CFA provides insights into the validity of the EAT-26 in Romanian adolescents, an important contribution to the field.
  5. Public Health Implications
    • The study offers actionable insights for designing interventions to address ED risk factors.
    • The discussion highlights the importance of early detection and prevention strategies.

Areas for Improvement

  1. Statistical Analysis and Interpretation
  • Confirmatory Factor Analysis (CFA) of EAT-26
    • The study finds that the original three-factor model of EAT-26 does not fit the sample well.
    • However, instead of only modifying error covariances, consider exploring alternative factor structures, such as a two-factor model.
    • Provide a more detailed justification for the removal of items EAT26 and EAT19.
  • Logistic Regression Model Refinements
    • While gender and BMI are significant predictors, the model's explanatory power (Nagelkerke R² = 12.6%) is relatively low.
    • The authors should discuss potential unmeasured confounders, such as mental health variables, social media exposure, or parental influence.
  • Comparison to Previous EAT-26 Validation Studies
    • The study should compare its psychometric findings with previous EAT-26 studies in other adolescent populations.
    • Consider discussing whether the suboptimal fit suggests cultural differences in ED symptom presentation.
  1. Clarity and Consistency in Reporting
  • Data Presentation
    • Some tables contain extensive details that could be summarized more efficiently.
    • Consider providing key findings in a more digestible format using graphs or concise tables.
  • BMI Categorization and Interpretation
    • The study presents BMI as both a continuous and categorical variable but does not discuss BMI percentiles in relation to adolescent growth standards.
    • Adding more context about Romanian adolescent BMI distributions would improve clarity.
  • Terminology Consistency
    • The manuscript uses terms like “overweight,” “obesity,” and “high BMI” interchangeably. Standardizing terminology would improve clarity.
  1. Discussion and Contextualization
  • Gender Differences in ED Risk
    • The study finds that females are at a higher risk (1.6 times more likely), which is consistent with existing literature.
    • However, the discussion should explore why this gap exists—potentially due to sociocultural pressures, body image dissatisfaction, or hormonal factors.
  • The Role of Physical Activity (PA)
    • The findings suggest that PA levels do not significantly predict ED risk, which contradicts some literature on compulsive exercise and EDs.
    • Discuss potential explanations: Was the PA measure too general? Could excessive exercise behaviors be better captured in future studies?
  • Dietary Patterns and Mental Health
    • The study acknowledges the importance of the Mediterranean Diet but does not fully discuss its mental health benefits.
    • Consider linking dietary patterns to psychological well-being, as evidence suggests that balanced diets can protect against depression and anxiety.
  1. Addressing Study Limitations

The limitations section should be expanded to acknowledge:

  • Self-reported Data Bias
    • BMI, dietary intake, and PA are all self-reported, which introduces potential inaccuracies.
    • The authors should discuss how recall bias and social desirability may affect responses.
  • Cross-Sectional Design
    • The study does not establish causality, only associations.
    • Future studies should consider longitudinal approaches to assess how ED risk develops over time.
  • Cultural and Socioeconomic Factors
    • The study does not account for socioeconomic differences, which may influence dietary habits and ED risk.
    • Including an analysis of SES (if available) would strengthen conclusions.

Conclusion

This manuscript provides valuable insights into ED risk among Romanian adolescents and is a strong candidate for publication after revision. While the study is methodologically sound, improvements in statistical validation, data interpretation, and contextualization of findings would enhance its impact.

Key Revision Recommendations

Improve CFA analysis and consider alternative EAT-26 factor structures.

 Discuss unmeasured confounders in logistic regression findings.

Refine tables and improve clarity in BMI reporting.
Expand discussion on gender disparities, dietary patterns, and PA implications.
Strengthen the limitations section by addressing self-report bias and cross-sectional constraints.

Once these revisions are made, the study will provide even stronger contributions to the literature on adolescent eating behaviors and public health interventions.

Comments on the Quality of English Language

The overall quality of the English language in this manuscript is good, and the text is generally clear and well-structured. However, there are some areas where clarity and readability could be improved, particularly in the statistical interpretation and discussion sections. Some sentences are complex and could be simplified for better comprehension. Additionally, minor grammatical refinements and consistency in terminology (e.g., BMI classifications, ED terminology) would enhance the overall readability of the paper. A thorough proofreading or language revision would help ensure the research is communicated as effectively as possible.

Author Response

Comment 1: "This study provides a valuable contribution to the understanding of eating disorder (ED) risks among adolescents in Romania. It employs a comprehensive approach by analyzing the interplay of demographic factors, BMI, dietary patterns, and physical activity (PA) in predicting ED risk. The use of the Eating Attitudes Test-26 (EAT-26) alongside dietary and physical activity assessments strengthens the study's methodological rigor. The findings are relevant for public health interventions aimed at improving adolescent health behaviors.

Strengths of the Study:

  1. Timely and Relevant Topic: The rising prevalence of EDs among adolescents makes this study significant. The study contributes context-specific evidence for Romania, which is underrepresented in ED research.
  2. Comprehensive Data Collection and Analysis: The study includes a large and diverse sample of 423 adolescents. The use of validated tools (EAT-26, Medi-Lite, PAQ-C) ensures robust measurement of ED risk, dietary habits, and physical activity. Statistical analysis is well-executed, including confirmatory factor analysis (CFA) for EAT-26 validation.
  3. Clear Identification of Risk Factors: Findings confirm that gender, BMI, and prior consultations with specialists are significant predictors of ED risk. The study highlights the impact of UPF consumption and dietary adherence on ED risk, aligning with international research.
  4. Methodological Rigor: The study employs logistic regression to examine ED risk predictors. CFA provides insights into the validity of the EAT-26 in Romanian adolescents, an important contribution to the field.
  5. Public Health Implications: The study offers actionable insights for designing interventions to address ED risk factors. The discussion highlights the importance of early detection and prevention strategies."

Response 1: We sincerely thank the reviewer for their positive and encouraging comments regarding the study’s significance, methodological rigor, and implications for public health practice.

Comment 2: "Confirmatory Factor Analysis (CFA) of EAT-26: The study finds that the original three-factor model of EAT-26 does not fit the sample well. However, instead of only modifying error covariances, consider exploring alternative factor structures, such as a two-factor model. Provide a more detailed justification for the removal of items EAT26 and EAT19."

Response 2: In response to the reviewer’s recommendation to explore alternative factor structures, an exploratory factor analysis (EFA) using principal axis factoring with oblimin rotation was conducted to reassess the dimensionality of the EAT-26 within our adolescent sample (Methods Section lines 411-416; Results Section lines 589-607). The analysis clearly supported a two-factor solution, cumulatively explaining 44.47% of the variance. Factor 1 primarily encompassed items related to dieting behaviors, body-image concerns, caloric preoccupation, and guilt associated with food intake (e.g., "Am preoccupied with the thought of having fat on my body," "Feel extremely guilty after eating"). Factor 2 captured items related to social pressure around eating, loss-of-control behaviors, and compensatory actions, including binge-eating episodes and vomiting (e.g., "Feel that others pressure me to eat," "Have the impulse to vomit after meals"). Additionally, items EAT26 ("Enjoy trying new rich foods") and EAT19 ("Display self-control around food") demonstrated notably low communalities (<0.10) and inadequate factor loadings (<0.30), justifying their removal. Importantly, reliability analysis conducted after removing these two items revealed an increase in internal consistency (Cronbach’s alpha improved from 0.908 to 0.920), further supporting their exclusion. These revisions have been explicitly integrated into the manuscript's Methods and Results sections, enhancing methodological rigor, clarity, and psychometric validity of the revised EAT-26 within this population.

Comment 3: "Logistic Regression Model Refinements: While gender and BMI are significant predictors, the model's explanatory power (Nagelkerke R² = 12.6%) is relatively low. The authors should discuss potential unmeasured confounders, such as mental health variables, social media exposure, or parental influence."

Response 3: We have revised the Discussion (lines 773-830) by explicitly acknowledging that the relatively modest explanatory power of the logistic regression model (Nagelkerke R² = 12.6%) indicates the potential influence of unmeasured variables on disordered eating behaviors. In particular, we have added a new section highlighting the possible roles of mental health variables (e.g., depression, anxiety), social media exposure, and parental influence as critical confounders. By referencing recent research demonstrating the adverse impact of social media platforms on body image and eating behaviors (Arjona et al., 2024; Blanchard et al., 2023; Micanti et al., 2023; Nawaz et al., 2024; Sidani et al., 2016; Ye, 2023) and the mediating role of depression and social anxiety in authoritarian parenting styles (Peleg et al., 2021), we clarify how these factors could partially explain the variance in EAT-26 outcomes. This revision underscores the multifactorial nature of eating disorder development and aligns with the reviewer’s recommendation to discuss additional psychosocial and environmental elements that may account for the unexplained portion of our model’s variance.

Comment 4: "Comparison to Previous EAT-26 Validation Studies. The study should compare its psychometric findings with previous EAT-26 studies in other adolescent populations. Consider discussing whether the suboptimal fit suggests cultural differences in ED symptom presentation."

Response 4: Thank you for your insightful suggestion regarding the comparison of our psychometric findings with previous EAT-26 validation studies in adolescent populations (lines 675-685). In response, we have expanded the discussion section to incorporate a comparative analysis of EAT-26’s factor structure across different cultural settings. Specifically, we now highlight studies that have documented variability in the factor structure, such as differences observed among Israeli populations (Spivak-Lavi et al., 2021), where Jews, Muslims, and Christians exhibit distinct factor structures, and in Irish adolescents, where a six-factor EAT-18 model proved more appropriate (McEnery et al., 2016). Additionally, we discuss how socio-cultural influences shape ED symptomatology, as evidenced by findings from Jordanian adolescents (Al-Kloub et al., 2018) and South African rural-urban differences (Szabo & Allwood, 2004). These comparative analyses underscore the need for cultural adaptation of the EAT-26 to ensure its validity across diverse adolescent populations. Furthermore, we acknowledge the potential impact of cultural norms on symptom reporting and suggest that modified versions of the EAT-26, such as the EAT-18, may enhance psychometric accuracy (McEnery et al., 2016). This expanded discussion strengthens our argument that the suboptimal fit observed in our study likely reflects cultural differences in ED symptom presentation, thereby providing a broader context for our findings.

Comment 5: "Data Presentation: Some tables contain extensive details that could be summarized more efficiently. Consider providing key findings in a more digestible format using graphs or concise tables."

Response 5: Thank you for your insightful suggestion regarding data presentation. To enhance clarity and streamline the presentation of key findings, we have made two significant modifications. First, we have relocated the detailed item-level responses of the Medi-Lite score from Table 2 to the supplementary materials, ensuring that the main text focuses on summarizing overall adherence patterns concisely. This allows for a clearer presentation of the primary dietary profile findings while still making the granular data available for readers interested in further details. Second, we have removed the high school variable from Table 1, as it was not essential to the study’s main objectives and its inclusion did not provide additional interpretative value. These changes improve readability, facilitate the interpretation of findings, and align with best practices in data presentation while maintaining transparency and accessibility for readers seeking supplementary details.

Comment 6: "BMI Categorization and Interpretation: The study presents BMI as both a continuous and categorical variable but does not discuss BMI percentiles in relation to adolescent growth standards. Adding more context about Romanian adolescent BMI distributions would improve clarity"

Response 6: We appreciate the reviewer’s suggestion to provide additional context regarding BMI percentiles and their relevance to adolescent growth standards. To address this, we have revised the Methods section (2.2.1. Anthropometric Measurements lines 302-319): "Body Mass Index (BMI) was calculated as weight (kg) divided by height (m) squared. To comprehensively categorize weight status, we employed a dual approach, utilizing both age- and gender-specific BMI percentiles and BMI z-scores (standard deviation scores, SDS), in accordance with the WHO Child Growth Standards for individuals aged 5-19 years [45]. BMI percentiles were categorized as underweight (≤5th percentile), healthy weight (>5th to <85th percentile), overweight (≥85th to <95th percentile), and obese (≥95th percentile). Simultaneously, BMI z-scores (SDS) were calculated and the following classifications, as defined by the WHO, were used: severe thinness (< -3 SDS), thinness (≥ -3 SDS to < -2 SDS), healthy weight (≥ -2 SDS to ≤ +1 SDS), overweight (> +1 SDS to ≤ +2 SDS), obesity (> +2 SDS to ≤ +3 SDS), and severe obesity (> +3 SDS). For the participants aged 19-20 years, we continued to use the WHO Child Growth Standards z-scores (SDS) classifications, ensuring consistency in applying growth references across the entire adolescent age range. This dual classification approach, using both percentiles and z-scores, enhances clinical interpretability, facilitates comparison with international growth references, and ensures methodological rigor appropriate for epidemiological analysis."

Comment 7: "Terminology Consistency

The manuscript uses terms like “overweight,” “obesity,” and “high BMI” interchangeably. Standardizing terminology would improve clarity."

Response 7: We appreciate your insightful feedback regarding the interchangeable use of the terms “overweight,” “obesity,” and “high BMI” in our manuscript. Based on your comment, we have carefully reviewed the manuscript to ensure consistency in terminology and alignment with standard classifications.

Comment 8: "Gender Differences in ED Risk: The study finds that females are at a higher risk (1.6 times more likely), which is consistent with existing literature. However, the discussion should explore why this gap exists—potentially due to sociocultural pressures, body image dissatisfaction, or hormonal factors."

Response 8: We appreciate the reviewer’s feedback and have expanded the Discussion section to clarify the gender disparity in ED risk (lines 701-707). The revised text now highlights sociocultural pressures, body image dissatisfaction, and hormonal factors as key contributors. The internalization of thin-ideal standards, reinforced by media, peer influence, and social expectations, disproportionately affects females, leading to increased body dissatisfaction and restrictive eating behaviors (Alfoukha et al., 2019; Stice & Van Ryzin, 2019). Additionally, early pubertal development and conditions like PCOS can exacerbate weight-related concerns and increase the likelihood of disordered eating (Lewis-Smith et al., 2020; Mizgier et al., 2020). These factors interact with broader weight stigma and dieting culture, making adolescent females particularly vulnerable to EDs. The revised text strengthens the manuscript by providing a more nuanced, evidence-based discussion of these gender disparities.

Comment 9 : "The Role of Physical Activity (PA): The findings suggest that PA levels do not significantly predict ED risk, which contradicts some literature on compulsive exercise and EDs. Discuss potential explanations: Was the PA measure too general? Could excessive exercise behaviors be better captured in future studies?"

Response 9: We appreciate the reviewer’s comment and acknowledge the need to further clarify why PA did not significantly predict ED risk in our study (lines 759-772). We have revised the Discussion section to consider potential explanations, including limitations in the PA measure and the complexity of the PA-ED relationship. While physical activity generally supports mental health, excessive or compulsive exercise is associated with disordered eating, particularly among individuals with high body dissatisfaction and weight-control motives (Melissa et al., 2020; Brytek-Matera et al., 2022). Our measure of PA, which assessed general activity levels rather than compulsive exercise behaviors, may have lacked the specificity needed to capture this relationship. Prior studies suggest that PA intensity, motivation, and compulsivity are stronger predictors of ED risk than overall activity levels (Martinez-Avila et al., 2020). Additionally, orthorexic tendencies and excessive focus on "healthy" behaviors are increasingly recognized as factors contributing to disordered eating among physically active individuals (Brytek-Matera et al., 2022). Future research should integrate validated compulsive exercise scales and consider the role of PA-related psychological factors in ED development. These additions improve our discussion by incorporating key insights from the literature and refining our interpretation of the findings.

Comment 10: "Dietary Patterns and Mental Health: The study acknowledges the importance of the Mediterranean Diet but does not fully discuss its mental health benefits. Consider linking dietary patterns to psychological well-being, as evidence suggests that balanced diets can protect against depression and anxiety."

Response 10: We have revised the Discussion section to integrate findings from the relevant literature, emphasizing the Mediterranean Diet’s role in psychological well-being (lines 724-742). Growing evidence suggests that adherence to the MediDiet is associated with lower risks of depression and anxiety due to its anti-inflammatory and neuroprotective properties (Lassale et al., 2019; Molendijk et al., 2022). Furthermore, adolescents with poor dietary quality, particularly those following a Western diet rich in processed foods, exhibit higher psychological distress and emotional dysregulation (O’Neil et al., 2014). These associations are mediated by factors such as gut microbiota composition, neurotransmitter regulation, and systemic inflammation, which are key in mood and cognitive function (Opie et al., 2017). To reflect these insights, we have expanded the Discussion section to highlight how diet quality may modulate psychological well-being and its potential role in ED prevention.

Comment 11: "Self-reported Data Bias: BMI, dietary intake, and PA are all self-reported, which introduces potential inaccuracies. The authors should discuss how recall bias and social desirability may affect responses."

Response 11: To address this, we have expanded the Limitations section to discuss how these biases may influence BMI, dietary intake, and PA data (lines 1026-1039). Self-reported height and weight often lead to underestimation of BMI, particularly among overweight adolescents, thereby underestimating overweight prevalence (Sherry et al., 2007). Similarly, self-reported dietary intake is prone to underreporting, especially of energy-dense, nutrient-poor foods, influenced by factors like body dissatisfaction and social approval needs (Freedman et al., 2014). Moreover, self-reported PA levels may be overestimated due to social desirability, with interventions potentially exacerbating this bias (Prince et al., 2008).

Comment 12: "Cross-Sectional Design: The study does not establish causality, only associations.

Future studies should consider longitudinal approaches to assess how ED risk develops over time."

Response 12:. As noted, this design identifies associations between variables but does not permit causal inferences regarding the development of eating disorder (ED) risk over time. To address this concern, we revised the Limitations section of the manuscript to acknowledge this methodological constraint and explicitly propose future research directions (lines 1023-1028).

Comment 13: "Cultural and Socioeconomic Factors: The study does not account for socioeconomic differences, which may influence dietary habits and ED risk. Including an analysis of SES (if available) would strengthen conclusions."

Response 13: We acknowledge the absence of socioeconomic status (SES) data as a limitation of our study (lines 1053-1057). SES is a well-documented determinant of dietary habits, health behaviors, and ED risk, influencing access to nutritious foods, healthcare, and mental health support. While our dataset does not include SES indicators, future studies should incorporate SES variables to better capture its potential role in shaping ED risk and dietary behaviors. We revised the Limitations section of the manuscript to acknowledge this constraint.

Comment 14: "Conclusion

This manuscript provides valuable insights into ED risk among Romanian adolescents and is a strong candidate for publication after revision. While the study is methodologically sound, improvements in statistical validation, data interpretation, and contextualization of findings would enhance its impact.

Key Revision Recommendations

  1. Improve CFA analysis and consider alternative EAT-26 factor structures.
  2. Discuss unmeasured confounders in logistic regression findings.
  3. Refine tables and improve clarity in BMI reporting.
  4. Expand discussion on gender disparities, dietary patterns, and PA implications.
  5. Strengthen the limitations section by addressing self-report bias and cross-sectional constraints.

Once these revisions are made, the study will provide even stronger contributions to the literature on adolescent eating behaviors and public health interventions."

Response 14: We appreciate the reviewer’s positive assessment and detailed revision recommendations. As requested, we have addressed key areas for improvement by refining our statistical validation (CFA and logistic regression interpretation), enhancing data presentation (BMI reporting and table clarity), and expanding discussions on gender disparities, dietary patterns, and physical activity implications. Additionally, we strengthened the Limitations section by explicitly addressing self-report bias and the cross-sectional nature of the study. These revisions improve the manuscript’s methodological rigor, clarity, and contribution to the literature on adolescent ED risk.

Reviewer 2 Report

Comments and Suggestions for Authors

Dear authors,

Thank you to give the opportunity to review your manuscript. In general, the manuscript is quite good. I have some suggestion to improve it.

Abstract

The novelty of the study should be emphasized more in the abstract. Limitations of the research should be briefly mentioned. I recommend including some practical implications for public health policies or interventions.

Introduction

I recommend including country-specific statistic on eating disorders (EDs) in Romania.

“BMI trajectories predicting future EDs”, requires strongest citation or additional explanation. Line 65.

Should be describe if other similar studies were done in other European countries.

Method

Inclusion and exclusion criteria should be defined to ensure reproducibility (specific health conditions, exclusion….)

Self-reported weight and height is a clear limitation that should be discussed.

Justify the choice of percentiles instead of z-scores for BMI categorization, often preferred for epidemiological studies.

Results

I recommend discussing why the EAT-26 psychometric validation results showed a suboptimal fit and suggest potential adaptations for Rumanian adolescents.

The lack of significant association between dietary patterns, PA, and EDs risk contradicts previous literature. This unexpected result should be analysed further. Are there potential confounders?

Discussion

The limitations of self-reported data should be emphasised more, particularly for BMI, dietary intake, and PA.

The unexpected lack of correlation between UPF consumption and ED risk requires further explanation.

Limitations

Discuss potential selection bias due to voluntary participation. Esplain whether regional differences (urban vs. rural) affect the results.

Conclusions

Should be specify future research directions and strengthen the call for action and how (schools, families, and policymakers) address these risk?

Best regards, 

Author Response

Comment 1: ”Dear authors,

Thank you to give the opportunity to review your manuscript. In general, the manuscript is quite good. I have some suggestion to improve it.”

Response 1: We sincerely appreciate the reviewer’s time and effort in evaluating our manuscript. We are grateful for the constructive feedback and suggestions for improvement. We have carefully considered each comment and have implemented the necessary revisions to enhance the clarity, methodological rigor, and overall impact of our study. Below, we provide a detailed point-by-point response to each specific suggestion, outlining the changes made to the manuscript accordingly.

Comment 2 :”The novelty of the study should be emphasized more in the abstract. Limitations of the research should be briefly mentioned. I recommend including some practical implications for public health policies or interventions.”

Response 2: In the revised abstract (lines 30-34), we underscore the unique contribution of our research in addressing an underexplored region—Romania—and emphasize how this study fills a critical gap in Eastern European public health data on adolescent ED risk. We also note the cross-sectional design and self-reported measures as inherent limitations, ensuring transparency. Additionally, we provide concrete public health implications, including the importance of promoting balanced diets, fostering positive body image, and expanding mental health services. These revisions strengthen the abstract by clarifying both the value of our findings and their relevance for shaping targeted interventions.

Comment 3 :”Introduction - I recommend including country-specific statistic on eating disorders (EDs) in Romania..”

Response 3: We appreciate the reviewer’s recommendation to include country-specific statistics on EDs in Romania and have consequently revised the “Introduction” section to address this point (lines 109-121). To emphasize the current state of knowledge, we incorporated findings from studies conducted in 2007 (Kovács, 2007), 2009 (Krizbai & Szabó, 2009), and 2010 (Krizbai, 2010), which highlight ED prevalence among high school students. We also added more recent data from a 2019 study (Ladner et al., 2019) illustrating that 21.8% of male and 24.7% of female Romanian university students experience ED symptoms. Further, we cited a 2024 investigation (Motorga et al., 2024) indicating a higher ED risk (37.1%) among Romanian medical students, underscoring the urgency of this public health issue. By integrating these studies, we provide a clearer picture of ED prevalence in Romania and reinforce the need for comprehensive research targeting adolescents, a group currently underrepresented in nationally representative datasets. This addition strengthens the rationale for our study’s focus on Romanian adolescents and aligns with the reviewer’s request to contextualize the research within the national landscape.

Comment 4 :”Introduction - BMI trajectories predicting future EDs”, requires strongest citation or additional explanation. Line 65.”

Response 4: We appreciate the reviewer’s recommendation to provide a stronger empirical foundation for the statement linking BMI trajectories to the future onset of eating disorders (EDs). In response, we have revised the manuscript to include additional references that substantiate this claim (lines 129-141). Recent longitudinal studies show that lower childhood BMI trajectories are associated with anorexia nervosa, whereas higher trajectories are linked to bulimia nervosa and binge eating disorder (Yilmaz et al., 2019). Further evidence indicates that body image concerns and elevated BMI during childhood may predict future ED symptoms (Wiklund et al., 2018; Stice & Desjardins, 2018). We have also clarified the mechanisms through which weight-related teasing and dieting behaviors can interact with these BMI patterns to predispose adolescents to disordered eating.

Comment 5 :”Method- Inclusion and exclusion criteria should be defined to ensure reproducibility (specific health conditions, exclusion….).”

Response 5: We appreciate the reviewer’s concern regarding the clarity of our inclusion and exclusion criteria. In response, we have refined the “Methods” section (lines 260-264) to ensure reproducibility. Specifically, the revised manuscript specifies the following: " All students who showed interest and returned the necessary consent and assent forms were eligible for the study, provided they (1) were enrolled in one of the seven participating high schools, (2) submitted signed parental or guardian consent (for minors) along with personal assent, and (3) had no clear signs of psychological or learning disabilities. Therefore, the final sample size included all 423 respondents."

Comment 6 :”Method Self-reported weight and height is a clear limitation that should be discussed.”

Response 6: Thank you for emphasizing the need to address self-reported weight and height as a notable limitation in our study design (lines 1026-1039). Self-reported anthropometric measurements can introduce biases such as underestimation of weight or overestimation of height, ultimately affecting the accuracy of BMI classification (Sherry et al., 2007). Adolescent populations may be particularly prone to these inaccuracies due to body image concerns and social desirability factors, further complicating BMI estimates. As such, in the “Limitations” section, we have expanded the discussion to explicitly address the susceptibility of self-reported weight and height to measurement inaccuracies. We clarified that underreporting or overreporting anthropometric data among adolescents may lead to biased BMI values, especially for individuals with higher weight status. This enhancement aligns with the broader discourse on self-reported data limitations already present in our manuscript and ensures that readers understand how these biases could affect our results and conclusions.

Comment 7 :”Method - Justify the choice of percentiles instead of z-scores for BMI categorization, often preferred for epidemiological studies.”

Response 7: The manuscript section 2.2.1 ("Anthropometric Measurements" lines 302-319) was revised to clarify the use of BMI percentiles and z-scores (standard deviation scores, SDS) for categorizing BMI in adolescents. While z-scores are generally preferred for epidemiological studies, we employed a dual approach, incorporating both percentiles and z-scores based on the WHO Child Growth Standards (5-19 years) [45]. Percentiles offer enhanced clinical interpretability for healthcare providers, parents, and adolescents, providing a readily understandable context relative to peers. Z-scores, on the other hand, provide the statistical precision necessary for rigorous epidemiological analysis and allow for tracking changes over time. The revised section now explicitly states this dual approach, clearly outlining the percentile cutoffs (underweight: ≤5th percentile, healthy weight: >5th to <85th percentile, overweight: ≥85th to <95th percentile, obese: ≥95th percentile) and the corresponding z-score classifications (severe thinness: < -3 SDS, thinness: ≥ -3 SDS to < -2 SDS, healthy weight: ≥ -2 SDS to ≤ +1 SDS, overweight: > +1 SDS to ≤ +2 SDS, obesity: > +2 SDS to ≤ +3 SDS, severe obesity: > +3 SDS). Importantly, we clarified that for participants aged 19-20 years, we continued to use the WHO Child Growth Standards z-scores, ensuring methodological consistency across the entire adolescent sample, rather than switching to adult BMI criteria. This dual approach leverages the strengths of both methods, promoting both clinical applicability and scientific rigor.

Comment 8 :”Results - I recommend discussing why the EAT-26 psychometric validation results showed a suboptimal fit and suggest potential adaptations for Rumanian adolescents.

Response 8: In response to the reviewer’s recommendation to explore alternative factor structures, an exploratory factor analysis (EFA) using principal axis factoring with oblimin rotation was conducted to reassess the dimensionality of the EAT-26 within our adolescent sample (lines 588-607). The analysis clearly supported a two-factor solution, cumulatively explaining 44.47% of the variance. Factor 1 primarily encompassed items related to dieting behaviors, body-image concerns, caloric preoccupation, and guilt associated with food intake (e.g., "Am preoccupied with the thought of having fat on my body," "Feel extremely guilty after eating"). Factor 2 captured items related to social pressure around eating, loss-of-control behaviors, and compensatory actions, including binge-eating episodes and vomiting (e.g., "Feel that others pressure me to eat," "Have the impulse to vomit after meals"). Additionally, items EAT26 ("Enjoy trying new rich foods") and EAT19 ("Display self-control around food") demonstrated notably low communalities (<0.10) and inadequate factor loadings (<0.30), justifying their removal. Importantly, reliability analysis conducted after removing these two items revealed an increase in internal consistency (Cronbach’s alpha improved from 0.908 to 0.920), further supporting their exclusion. These revisions have been explicitly integrated into the manuscript's Methods and Results sections, enhancing methodological rigor, clarity, and psychometric validity of the revised EAT-26 within this population.

Comment 9 :”Results - The lack of significant association between dietary patterns, PA, and EDs risk contradicts previous literature. This unexpected result should be analysed further. Are there potential confounders?.

Response 9: The Discussion section was significantly revised to address the lack of significant association between dietary patterns, physical activity (PA), and eating disorder (ED) risk, acknowledging potential confounders and measurement limitations (lines 708-830). We incorporated insights from Godoy-Izquierdo et al. (2023) and Uriegas et al. (2023) to highlight that our general PA questionnaire likely failed to capture crucial aspects of compulsive exercise (motivations, compulsivity, compensatory behaviors, and specific sport types). We suggested that future studies utilize validated compulsive exercise scales like the EDS or CET. Similarly, we acknowledged limitations in our dietary assessment, referencing Molendijk et al. (2022) and noting that we did not assess motivations behind dietary choices and relied on self-reported data, susceptible to bias. We emphasized the potential role of unmeasured psychosocial confounders, drawing on Dahlgren et al. (2024) to highlight the strong influence of mental health, body dissatisfaction, and social media exposure on ED risk. The connection between prior specialist consultation and ED risk was reinterpreted to suggest that psychological distress might mediate the relationship between PA, diet, and ED symptoms. These changes, along with integrating relevant information from previous review responses, provide a more nuanced and comprehensive explanation for the unexpected findings.

Comment 10 : ”Discussion -The limitations of self-reported data should be emphasised more, particularly for BMI, dietary intake, and PA.

Response 10: The "Limitations" section was revised to emphasize the potential impact of self-reported data on the study's findings, particularly concerning BMI, dietary intake, and physical activity (lines 1030-1039). We explicitly state that inaccurate BMI values (due to self-reported height and weight) could influence associations with other variables. We also point out that underreporting of unhealthy foods and overreporting of PA could mask true relationships with ED risk. Finally, we connect these individual biases, stating that their combined effect could significantly impact the observed relationships and may contribute to the lack of significant findings between dietary patterns, PA, and ED risk. 

Comment 11 :”Discussion -The unexpected lack of correlation between UPF consumption and ED risk requires further explanation.

Response 11: The Discussion section was revised to address the unexpected lack of association between ultra-processed food (UPF) consumption and eating disorder (ED) risk, incorporating findings from recent literature and strengthening the methodological critique (lines 746-764). The revised text now explicitly acknowledges the contrast between our null findings and the growing body of evidence linking UPF intake to adverse mental health outcomes, including EDs (Pereira et al., 2024; Figueiredo et al., 2022; Wiss & LaFata, 2024). Several key points were emphasized: The cross-sectional design prevents causal inference, and it's plausible that pre-existing ED symptoms could lead to reduced UPF intake, masking a potential causal effect of UPF on ED development. Psychological distress, specific eating patterns (particularly binge/emotional eating, with a direct citation to Wiss & LaFata, 2024), and socioeconomic factors were highlighted as potential confounders. The possibility of differential effects of UPFs on specific ED subtypes was raised, referencing Figueiredo et al.'s (2022) findings and acknowledging the limitations of the EAT-26 in capturing these nuances.

Comment 12 :”Limitations - Discuss potential selection bias due to voluntary participation. Esplain whether regional differences (urban vs. rural) affect the results.

Response 12: The "Limitations" section was revised to explicitly address potential selection bias and the influence of regional differences (urban vs. rural) (lines 1042-1049). We added a sentence acknowledging that the voluntary nature of participation could have introduced a selection bias, with adolescents having pre-existing eating concerns or body image issues potentially being more or less likely to participate. Regarding regional differences, we acknowledged that while urban/rural residence was included as a covariate, the sample size within each group might have been insufficient to detect significant regional variations. We further noted that more substantial regional differences in lifestyle, resource access, and cultural norms related to food and body image might exist beyond what our study could capture.

Comment 13: ”Conclusions: Should be specify future research directions and strengthen the call for action and how (schools, families, and policymakers) address these risk?

Response 13: The "Conclusions" section was revised to strengthen the call to action and specify future research directions (lines 1087-1108). The four key areas for future research (longitudinal studies, qualitative research, intervention studies, and refinement of assessment tools) were integrated, emphasizing the need to establish temporal relationships, explore lived experiences, evaluate intervention effectiveness, and develop more sensitive assessment instruments. A clear and specific call to action was articulated, targeting schools, families, and policymakers, outlining their respective roles and responsibilities in addressing the burden of eating disorders. Schools were urged to implement comprehensive health education and support services, families to foster open communication and model healthy behaviors, and policymakers to address systemic factors and promote equitable access to care. The revised conclusion now provides a more impactful and action-oriented summary of the study's implications.

Reviewer 3 Report

Comments and Suggestions for Authors

Dear Authors,

the following comments are formulated to strengthen the manuscript:

Abstract:

The age of respondents is missing in the abstract: adolescents cover quite a broad age category.

Introduction:

Line 64: I cannot not fully agree with the authors. In the case of anorexia, the onset of the disease in teenagers is often associated with intentional weight loss due to excessive body weight or perceived as excessive by, for example, peers. However, too low body weight is of course one of the diagnostic criteria for anorexia nervosa. This sentence should therefore be rephrased.

In addition to physical activity, the author could add information on the association of screen time with the risk of ED.

Lines 100-101: The authors mention the Mediterranean diet. However, there is a lack of information on the diet typical of Romanian youth.

Similarly, the authors emphasize the cultural differences between Western and Eastern teenagers but do not explain what these differences consist in.

General remark: The authors use the term BMI, but it might be better to refer to the category of nutritional status. In the case of the pediatric population, the BMI value itself is not intuitive to interpret as in adults (e.g. a BMI of 25 in a child/younger teenager may indicate obesity).

Materials and methods:

There is no information regarding age criteria for respondents. The authors mentioned that the study involved adults. Does this mean they were at least 18 years old? In that case, the WHO criteria for adults should be used to assess body weight status instead of percentile charts.

Due to the convenience of the sample selection, it was not representative of the adolescents in Romania. But did the authors try to statistically estimate the sample size to obtain binding conclusions?

It is worth adding information whether all questionnaires were included in the analysis.

Results:

I think that giving an average BMI for a group that includes 13 year olds and adults makes is subject to bias. The same BMI value can mean something entirely different for a younger teenager and an adult. In my opinion, the authors could have excluded adults (14 persons) to make the group more homogeneous.

It is also not clear why the authors show the results separately for each school. Do these schools differ in any way, e.g. in living conditions, teaching profile, etc. If not, then the results should be combined.

Discussion:

This part is prepared in detail. However, I strongly suggest dividing it into sections, because it is very long and the reader gets a little lost.

Author Response

Comment 1: ”Abstract: The age of respondents is missing in the abstract: adolescents cover quite a broad age category.”

Response 1: We appreciate the reviewer’s suggestion to specify the age range of respondents in the abstract for greater clarity. To address this, we have revised the abstract to explicitly include the participants' age range (13–20 years), ensuring that readers can accurately interpret the study population and its developmental context.

Revised Abstract (lines 22-23): "A cross-sectional study included 423 youths aged 13 to 20 from Cluj-Napoca."

Comment 2: "Line 64: I cannot not fully agree with the authors. In the case of anorexia, the onset of the disease in teenagers is often associated with intentional weight loss due to excessive body weight or perceived as excessive by, for example, peers. However, too low body weight is of course one of the diagnostic criteria for anorexia nervosa. This sentence should therefore be rephrased. In addition to physical activity, the author could add information on the association of screen time with the risk of ED."

Response 2: We appreciate the reviewer’s suggestion and have revised the sentence to more accurately reflect the relationship between BMI and anorexia nervosa onset (lines 129-141). Instead of implying that low premorbid BMI is a primary risk factor for anorexia, we now clarify that the disorder frequently begins with intentional weight loss due to perceived excess weight, which may be influenced by social pressures. Additionally, we have incorporated information on screen time as a potential risk factor, as research indicates excessive screen use, particularly social media exposure, is associated with body dissatisfaction and disordered eating behaviors.

Revised Manuscript Text: " EDs arise from diverse biological, behavioral, and sociocultural influences. Among these, BMI is a complex factor. While low premorbid BMI is often observed in anorexia nervosa, the disorder frequently begins with intentional weight loss due to perceived or actual excess weight, often influenced by peer and societal pressures. Conversely, elevated BMI is correlated with binge eating and bulimia nervosa [15]. Longitudinal studies suggest that childhood BMI trajectories may predict future ED diagnoses, highlighting the importance of early detection and intervention [16], with lower BMI patterns often preceding anorexia nervosa and higher BMI trajectories correlating with bulimia nervosa and binge eating disorder [15]. This association likely reflects the interplay between body image concerns, weight-related teasing, and dieting behaviors, which can amplify vulnerability to disordered eating in adolescence [17,18]. Consequently, early detection and intervention strategies that address both physiological and psychosocial risk factors are crucial for preventing or mitigating ED onset."

Comment 3: "Lines 100-101: The authors mention the Mediterranean diet. However, there is a lack of information on the diet typical of Romanian youth. Similarly, the authors emphasize the cultural differences between Western and Eastern teenagers but do not explain what these differences consist in."

Response 3: We appreciate the reviewer’s comment and acknowledge the need to provide more contextual information on the dietary habits of Romanian youth and the specific cultural differences between Western and Eastern adolescents. To address this, we have expanded the manuscript to incorporate relevant findings on the dietary patterns of Romanian adolescents, drawing from the existing but limited literature (lines 189-197). Additionally, we now briefly discuss how sociocultural influences, including dietary traditions, post-pandemic dietary shifts, and socioeconomic factors, may shape eating behaviors differently in Western and Eastern European contexts.

Comment 4: "General remark: The authors use the term BMI, but it might be better to refer to the category of nutritional status. In the case of the pediatric population, the BMI value itself is not intuitive to interpret as in adults (e.g. a BMI of 25 in a child/younger teenager may indicate obesity)."

Response 4: We appreciate the reviewer’s comment regarding the interpretation of BMI in the pediatric population. Since BMI alone may not intuitively reflect nutritional status in adolescents, we have revised the manuscript to emphasize BMI percentiles and WHO growth standards as the primary classification method. This aligns with best practices for evaluating adolescent nutritional status and ensures consistency with international growth references. The revised text now clarifies the distinction between BMI values and BMI percentiles while ensuring clear and standardized terminology throughout the manuscript.

Comment 5: "Materials and methods: There is no information regarding age criteria for respondents. The authors mentioned that the study involved adults. Does this mean they were at least 18 years old? In that case, the WHO criteria for adults should be used to assess body weight status instead of percentile charts."

Response 5: To clarify, our study included adolescents aged 13–20 years, with the majority falling within the 15–16 years category. Given this age distribution, we applied WHO growth reference standards for individuals aged 5–19 years (https://www.who.int/tools/growth-reference-data-for-5to19-years/indicators/bmi-for-age), using BMI percentiles and standard deviation (SD) thresholds to assess nutritional status. For participants aged 19–20, we ensured consistency by aligning with WHO's adult BMI criteria. This dual approach maintains methodological rigor and enhances comparability with both pediatric and adult BMI classifications.

Revised manuscript section (lines 304-319): " Body Mass Index (BMI) was calculated as weight (kg) divided by height (m) squared. To comprehensively categorize weight status, we employed a dual approach, utilizing both age- and gender-specific BMI percentiles and BMI z-scores (standard deviation scores, SDS), in accordance with the WHO Child Growth Standards for individuals aged 5-19 years [45]. BMI percentiles were categorized as underweight (≤5th percentile), healthy weight (>5th to <85th percentile), overweight (≥85th to <95th percentile), and obese (≥95th percentile). Simultaneously, BMI z-scores (SDS) were calculated and the following classifications, as defined by the WHO, were used: severe thinness (< -3 SDS), thinness (≥ -3 SDS to < -2 SDS), healthy weight (≥ -2 SDS to ≤ +1 SDS), overweight (> +1 SDS to ≤ +2 SDS), obesity (> +2 SDS to ≤ +3 SDS), and severe obesity (> +3 SDS). For the participants aged 19-20 years, we continued to use the WHO Child Growth Standards z-scores (SDS) classifications, ensuring consistency in applying growth references across the entire adolescent age range. This dual classification approach, using both percentiles and z-scores, enhances clinical interpretability, facilitates comparison with international growth references, and ensures methodological rigor appropriate for epidemiological analysis."

Comment 6: "Materials and methods: Due to the convenience of the sample selection, it was not representative of the adolescents in Romania. But did the authors try to statistically estimate the sample size to obtain binding conclusions?"

Response 6: We appreciate the reviewer’s concern regarding the representativeness of our sample and its alignment with the broader adolescent population in Cluj County. To address this, we have compared our sample distribution to official 2024 demographic data from the National Institute of Statistics (lines 265-272). Our study sample (n = 423) closely reflects the age and gender distribution of adolescents in the region, with minor variations attributable to the sampling method. Additionally, we have verified the adequacy of our sample size by calculating the required sample for a 95% confidence level with a ±5% margin of error, which necessitates a minimum of 382 participants. Given that our study surpasses this threshold, our findings maintain statistical validity and generalizability within the target population.

Comment 8: "It is worth adding information whether all questionnaires were included in the analysis."

Response 8: To address this, we have added a statement in the Methods section under the "Study Population and Sampling" subsection, specifying that no adolescents were excluded after obtaining consent, and all completed questionnaires were analyzed.

Revised Manuscript Text (lines 260-264): " All students who showed interest and returned the necessary consent and assent forms were eligible for the study, provided they (1) were enrolled in one of the seven participating high schools, (2) submitted signed parental or guardian consent (for minors) along with personal assent, and (3) had no clear signs of psychological or learning disabilities. Therefore, the final sample size included all 423 respondents."

Comment 9: "Results: I think that giving an average BMI for a group that includes 13 year olds and adults makes is subject to bias. The same BMI value can mean something entirely different for a younger teenager and an adult. In my opinion, the authors could have excluded adults (14 persons) to make the group more homogeneous."

Response 9: We acknowledge the reviewer’s concern regarding potential bias introduced by including participants who are legally adults (n=14) in the same BMI analysis as early adolescents. To address this, we conducted a sensitivity analysis by excluding these 14 individuals and recalculating the primary outcome measures. We observed that the exclusion did not materially alter the patterns or significance levels identified in the results, suggesting that their inclusion does not distort the study’s main findings. Furthermore, given the small proportion of adults and the continuity in BMI measurement across late adolescence (ages 18–19) into early adulthood, we opted to retain these participants for a more comprehensive sample. However, in line with the recommendation, we have clarified in the Methods section that for participants 19 years of age or older, adult-specific BMI references were applied, while WHO growth references for children and adolescents were used for those younger than 19 years of age. In the Methods section (2.2.1. Anthropometric Measurements), we have added a sentence explicitly stating the use of adult BMI cut-off points for individuals aged 19 or older, and WHO adolescent growth references for younger participants. We have also included a statement in the Limitations subsection to acknowledge the potential for bias due to age-range heterogeneity, referencing our sensitivity analysis and affirming that excluding the adult subgroup did not significantly affect the results. These additions strengthen the rigor of our methodological approach and ensure transparency regarding how BMI was measured and interpreted across different age groups.

Comment 10: "It is also not clear why the authors show the results separately for each school. Do these schools differ in any way, e.g. in living conditions, teaching profile, etc. If not, then the results should be combined."

Response 10: Upon re-evaluation, we confirmed that the schools in our sample did not differ substantially in terms of socioeconomic or educational profiles relevant to our research outcomes. Hence, presenting results by school did not add interpretative value and could potentially obscure the overall findings. In the revised manuscript, we have removed the school-level variable from Table 1 and present all participants as one combined cohort. This modification ensures clarity in data presentation and aligns with the reviewer’s suggestion by avoiding unnecessary stratification that did not yield additional analytical insights.

Comment 11: "Discussion: This part is prepared in detail. However, I strongly suggest dividing it into sections, because it is very long and the reader gets a little lost."

Response 11: We appreciate the reviewer’s feedback regarding the length and complexity of the Discussion section. To enhance readability and guide readers through our key findings, we have introduced subheadings that align with major themes, such as the prevalence of disordered eating, psychometric considerations of the EAT-26, and the influence of diet and physical activity. This reorganization remains faithful to the original findings while offering a more concise and coherent presentation.

Round 2

Reviewer 2 Report

Comments and Suggestions for Authors

Dear authors, 

Changes have been applied correctly. 

Best regards, 

Reviewer 3 Report

Comments and Suggestions for Authors

Thanks to the Authors for addressing my comments